# V-CECE: Visual Counterfactual Explanations via Conceptual Edits

**Nikolaos Spanos** **Maria Lymperaiou** **Giorgos Filandrianos**
**Konstantinos Thomas** **Athanasios Voulodimos** **Giorgos Stamou**
National Technical University of Athens
{nspanos,marialymp,geofila,kthomas}@ails.ece.ntua.gr
thanosv@mail.ntua.gr, gstam@cs.ntua.gr

## Abstract

Recent black-box counterfactual generation frameworks fail to take into account the semantic content of the proposed edits, while relying heavily on training to guide the generation process. We propose a novel, plug-and-play black-box counterfactual generation framework, which suggests step-by-step edits based on theoretical guarantees of optimal edits to produce human-level counterfactual explanations with zero training. Our framework utilizes a pre-trained image editing diffusion model, and operates without access to the internals of the classifier, leading to an explainable counterfactual generation process. Throughout our experimentation, we showcase the explanatory gap between human reasoning and neural model behavior by utilizing both Convolutional Neural Network (CNN), Vision Transformer (ViT) and Large Vision Language Model (LVLM) classifiers, substantiated through a comprehensive human evaluation. Project page and code are available at https://nickspanos55.github.io/vcece

## 1 Introduction

Ensuring fairness and trust in artificial intelligence (AI) applications has been of paramount importance, especially since the vast adoption of large and uninterpretable models, such as Large Language Models (LLMs) [33] and Diffusion Models [17]. For this reason, explainable AI (XAI) has become an essential research field, fostering understanding and accountability of black-box models, while rendering them ethically deployable in practical high-stakes circumstances [35]. Among the various XAI techniques, counterfactual explanations (CEs) stand out as a powerful tool for providing actionable and human-centric insights [14]. The *what-if* nature of CEs underpins cause-effect relationships in human reasoning [18], aligning with the concept of performing minimal input perturbations to simulate those *what-if* scenarios. Observable outcome changes indicate that the corresponding input perturbation, though subtle, was sufficiently influential, uncovering a reasoning path inside the model. By aggregating these reasoning paths within a well-structured CE framework, general patterns of the model's decision-making are revealed, highlighting potentially flawed reasoning directions.

The parallel venture of counterfactual generation considers a generated image $x^*$ as an imaginary counterpart of an existing image $x$ which succeeds in altering the prediction of a classifier $C$. To this end, an appropriate CE framework should be able to answer *why $x^*$* was classified in a class $L^*$ rather than $L \neq L^*$ in human-interpretable terms. The real value of CEs in comparison to any possible perturbation that achieves flipping from $L$ to $L^*$ lies in the *semantics* [5] considered for classification, as well as for the generation of $x^*$: high-level concepts succeed in explaining *why $L^*$ instead of $L$*, while superficial changes –such as altering a pixel in an image– may flip the classification label but fail to provide a meaningful or tractable explanation for the reason behind the change.

39th Conference on Neural Information Processing Systems (NeurIPS 2025).

The counterfactual image generation literature frequently reports dispersed edits [3, 25, 32, 23], non-actionable outcomes [36] or uninterpretable changes overall [19, 10, 42, 22] all of which do not support the semantic desiderata of CEs in the first place. Even though high-quality generations are often achieved, they are not reproducible or interpretable by humans [21], undermining the primary goal of enhancing human understanding of model decision-making [5, 39]. Additionally, methods that apply edits to generate images without user explanations, such as those relying on diffusion models trained with classifier labels [23, 21], can yield misleading results due to their heavy reliance on the diffusion process. For example, Stable Diffusion may add a blob resembling a bus, leading to the mistaken conclusion that the bus influences the classifier's output, while it is unclear if the classifier indeed identifies the bus or reacts to pixel distribution changes.

At the same time, semantic-driven CE algorithms underline biases of black-box models [8, 9], ensuring actionability and interpretability of edits under strict explanation frameworks. However, they assume a-priori that the classifier under explanation holds the same understanding of semantics as humans. To this end, we observe a significant gap in CE literature: there is no investigation on what characteristics are important for CEs from the perspective of neural classifiers and humans, and where these two perspectives disagree. In this paper, we argue that this issue is even more problematic than the case of uninterpretable adversarial edits, as it introduces ambiguity regarding whether the edits are comprehensible and reproducible for humans. This ambiguity can result in misleading CEs, ultimately compromising the effectiveness of related explainability algorithms. By acknowledging these limitations, we break down the problem of defining CEs into two discrete steps. The first one addresses the question: *"Is there a discrepancy between human and neural perspectives?"*. In other words, *"Can the classifier's decision-making process be explained using human-level semantics?"*. If the answer to the first question is affirmative, the second step is to determine: *"What are the minimum edits that actually change the classifier's label?"*

Existing counterfactual frameworks often address the second question only partially, either bypassing the first question entirely or assuming an ungrounded answer. Works such as [23, 21] disregard semantics entirely, making it difficult to accurately interpret the second question, as the context provided by the first is essential for understanding its implications. On the other hand, approaches that assume the classifier operates at a semantic level [9, 8] fail to provide evidence or indications to substantiate this assumption, effectively sidestepping the first question altogether.

Our proposed V-CECE is the first to systematically address *both of these questions*, introducing a CE generation pipeline focused on two main directions: first, we exclusively support *conceptual edits* based on well-defined visual semantics, guaranteeing the optimality of edits. Secondly, we apply the extracted optimal edits using *state-of-the-art (SoTA) diffusion models* for counterfactual image generation. Ultimately, the effectiveness of these images is used as a proxy for identifying the *discrepancy between the classifier's and human perspectives*. Overall, our generation pipeline operates in a *fully black-box setting* in which we do *not* train any of the participating modules, nor do we optimize over the final CEs, offering a highly efficient plug-and-play solution.

## 2   Related work

**Counterfactual explanations**   comprise a human-interpretable way of explaining deep learning models, applicable in several scenarios [14], thus they have been favored in recent explainability literature. By probing a pre-trained model and observing its output changes, we can approximate its behavior without accessing its internals [40], a requirement that is on par with the development of proprietary models. A line of work focuses on concept-based counterfactuals [13, 2, 1, 39, 8, 9], driven by the claim that there can be no explanation without semantics [5]. In that case, higher-level concepts are used to explain *why* a classifier made a decision and what *could have been* different in order to alter the classification outcome. In this paper, we exclusively work with conceptual CEs.

**Diffusion models for counterfactuals**   The advent of diffusion models [17] has rapidly elevated the field of image synthesis, also allowing high-quality counterfactual image generation. Initial attempts manage to modify observable regions of an image to change its classification [3], even though they do not deal with well-defined semantics. Causal white-box frameworks enhance counterfactual generation with theoretical constrains [36], even though their actual generations alter class identity in somehow non-realistic directions. Also in the white-box spectrum [7, 21, 22, 37, 25, 32], access to the classifier's gradients is required to generate counterfactuals, thus being unable to explain

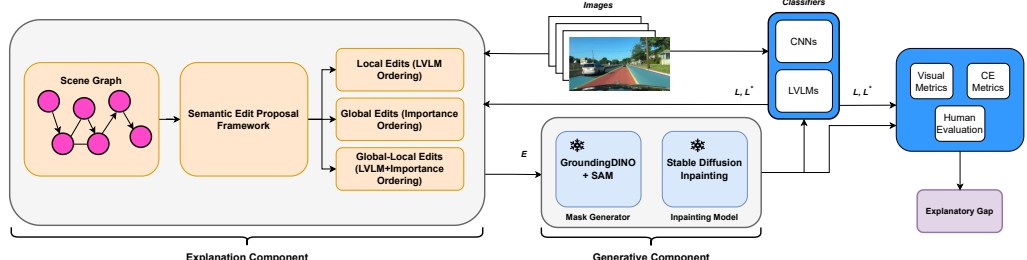

Figure 1: Outline of V-CECE to address the explanatory gap between humans and models. The semantic edit framework reviews the image and proposes edits and their iterative sequence. The edits are then implemented through a combined object recognition and diffusion model. The edited images are reviewed from the respective models to ascertain whether or not the edit had the desired effect. The edited images are evaluated through visual metrics, counterfactual metrics and a human survey.

proprietary classifiers. [10] deviate towards model-agnostic approaches, even though they do not focus on high-level concepts, while misgenerations are also present. In the black-box setting also lies the work of [23], producing dispersed edits within the images. Other endeavors target to manipulate the presence or absence of features rather than flipping the classification label [42]. Our work belongs in the model-agnostic setting, treating *both* the classifier as well as the diffusion model as black boxes, while only human-interpretable conceptual edits are considered over other perturbations.

# 3  Method

Our V-CECE pipeline mainly consists of an explanation component and a generative component (Figure 1). In the explanation stage, we query for the closest image pair in terms of semantics that belong to distinct visual classes $L, L^*$, given any pre-trained classifier $C$. To optimally transit from $L$ to $L^*$, a set of conceptual edits $E$ is calculated, incorporating feasible insertions $I$, deletions $D$ and substitutions $S$ of concepts. The guarantee of optimality of the proposed method stems from the underlying mechanism used to compute the closest image pair, as well as the correspondingly optimal calculation of the edit set $E$. Once computed, $E$ is passed to the generative stage which ultimately executes them to produce the counterfactual image. A grounding module masks the area to be edited, driving the generation process of a diffusion model. In each generation step, $C$ decides whether the generated image actually changed its class or not, terminating the generation process in the first case.

## 3.1  Guarantees of optimality

The explanation component is responsible for guaranteeing that the proposed semantic explanations to achieve $L \rightarrow L^*$ are optimal, meaningful and actionable. Specifically, it suggests insertions $I$, deletions $D$ and replacements $R$ of concepts driven by their distances on a pre-defined knowledge graph. In our work, we utilize WordNet [31], due to its inclusion of multiple semantic meanings and its presence in previous work, such as in [8, 9, 11]. Given a semantic concept $s_i$ from a sample $x_i \in L$ and another semantic concept $s_j$ from a sample $x_j^* \in L^*$ ($s_i \neq s_j^*$), the cost $c(S_{s_i \rightarrow s_j^*})$ of substituting optimally $s_i$ with $s_j$ is equivalent to finding the shortest path $min(dist(s_i, s_j^*))$ between these concepts on the knowledge graph using pathfinding algorithms, such as Dijkstra. Similarly, $I$ and $D$ operations require traversing the knowledge graph up to its root, invoking a related edit cost ($c(I_{s_j^*})$ for $I$ and $c(D_{s_i})$ for $D$) equal to the distance of this path. Actionability of edits is also guaranteed, as non-actionable edits can be excluded by assigning them an infinite cost, thereby removing them from consideration. Overall, the edits to be performed in each step of the $L \rightarrow L^*$ transition are defined by the following optimization function:

$$min(\sum_{s_j^* \in L^*} c(I_{s_j^*}), \sum_{s_i \in L} c(D_{s_i}), \sum_{s_i \in L, s_j^* \in L^*} c(S_{s_i \rightarrow s_j^*})) \qquad (1)$$

The solution of equation 1 can be deterministically provided using bipartite matching, where concepts from $L$ and $L^*$ items are placed onto a bipartite graph $\mathcal{G}$, with each edge $e_{(s_i, s_j^*)}$ having corresponding weight $w_{e_{(s_i, s_j^*)}} = c(S_{s_i \rightarrow s_j^*})$, while also dummy nodes $d_{s_i}, i_{s_j^*}$ to simulate $I$ and $D$ operations are

added, with associated costs of $w_{e_{(s_i, d_{s_i})}} = c(D_{s_i}), w_{e_{(s_j^*, i_{s_j^*})}} = c(I_{s_j^*})$. The minimization of this matching is performed via the Hungarian algorithm [26], resulting in a set $E = \{I, D, S\}$ of minimum cost edits that satisfy the $L \to L^*$ transformation. Bipartite matching is analogous to an $m \times n$ assignment problem, where $m$ and $n$ correspond to the number of source and target concepts respectively; this problem is solvable in $O(mn \log n)$ time via the Hungarian algorithm.

## 3.2 Selection of edits

The explanation component returns the optimal set of semantic edits $E$ to transform an input image belonging to class $L$ into an existing image classified as $L^*$. Importantly, $E$ constitutes the *provably minimal* set of edits such that, if all are applied, the predicted class is guaranteed to change. However, there is no guarantee that this set is *uniquely minimal*, nor that a proper subset $E' \subset E$ could not also suffice to induce the same class change. To approximate such a minimal effective subset $E'$, we iteratively apply edits from $E$ until the classifier's prediction transitions from $L$ to $L^*$. s To determine which edits $e_{(s_i, s_j^*)} \in E$ will be actually performed, we employ three distinct methods.

**Local Edits**  After extracting the edits $e_{(s_i, s_j^*)} \in E$ as driven from the explanation component, we must decide their order. Lacking human context cues, we delegate this ordering procedure to a Large Vision–Language Model (LVLM). At every step, the LVLM receives the current image plus the remaining edits from $E$ and selects the next action–*insert*, *delete*, or *subsitute*. We then update the image and repeat. Supplying the updated image each round prevents the logical inconsistencies that arise when the whole sequence is produced at once. Prompts are provided in App. **B**.

**Global Edits**  Local reasoning overlooks systematic biases present in the classifier. Inspired by [8], we ask: *Which semantic edits most often flip images from class $L$ to $L^*$?*. Running the Section 3.1 algorithm over all images in $L$, we tally every edit $e_{(s_i, s_j^*)}$ and score it according to the following:

$$\text{Importance}(e_{(s_i, s_j^*)}) = \frac{|I_{s_j^*}| - |D_{s_i}| + |S_{s_i \to s_j^*}| - |S_{s_j^* \to s_i}|}{|e_{(s_i, s_j^*)} \in E|}. \tag{2}$$

We then apply edits in descending importance: e.g. if *delete bed* ranks highest and a bed is present, we remove it, test if the label is altered, or proceed to the next edit until the class changes.

**Local–Global Edits**  Global ordering exploits classifier shortcuts based on global biases but ignores scene details, whereas local ordering does the opposite. A balanced approach applies only the local edits proposed for the specific image, yet orders them by the global importance scores, thus retaining scene-aware changes while still taking advantage of biases imbued in the classifier.

## 3.3 Performing the Edits

We apply edits to the image using a pre-trained, frozen diffusion model. While training the diffusion model could improve the alignment of generated images with the dataset, it would also transfer statistical biases present in the data to the generation process [12, 15], leading to artificially favorable counterfactual images; this runs counter to our goal of using semantic edits to test the classifiers' semantic comprehension. By relying on a model not directly trained on our data, we ensure that any inherent bias remains consistent across all experiments, creating a fair foundation for comparison.

We leverage the Stable Diffusion v1.5 Inpainting model [1] to edit images while closely following the prompts and fully repainting the masked regions. Each image is processed for 40 steps with the DPM++ 2M SDE sampler [34] and an automatically selected scheduler. We opt for the default random seed, which can be fixed for reproducibility, while we abstain from applying a variation seed to keep outputs consistent. A high-resolution fix is enabled, adding an extra upscaling pass that improves final image quality.

To minimize commonsense artifacts introduced during editing, additional information should be provided. This information includes the optimal placement within the image for any object that has to be added. For instance, if a pillow is to be inserted, the most suitable location for it must be

---

[1]Model card: ruwnayml/stable-diffusion-inpainting

determined, such as on a couch. Moreover, when removing an object, it is pertinent to consider what is most likely to be behind it and to replace it accordingly in order to maintain image continuity. Both of these steps are performed using the reasoning and common-sense understanding capabilities of a LVLM [44, 6], and specifically Claude 3.5 Sonnet, which processes the image along with a task-specific prompt (to add or remove an object) as input. More details can be found in App. **B**.

## 4 Experiments

**Datasets**   We experiment with distinct datasets for which the semantics play a definitive role. First, we utilize BDD100K [43] that focuses on real-world autonomous driving situations, where semantics are important for whether a car has to stop or move. This dataset has been favored in previous counterfactual generation works [21, 23] thanks to the well-defined semantics representing each class. Moreover, following the state-of-the-art work on semantic counterfactuals [9], we replicate the Visual Genome experiments on the VG-Random subset[2], upon which we generate the final images. The object annotations of each image and dataset comprise the concept sets considered in the bipartite graph construction of the explanation component.

**Classifiers**   To ensure a fair comparison of our results with other methods for the BDD100K dataset, we employ the same DenseNet-121 classifier as in [23]. We extend to more convolutional classifiers, and specifically ConvNext [30] and EfficientNet [38], as well as to transformer-based architectures, such as Swin [29]. Similarly, for Visual Genome, we use the ResNet18 classifier from [8, 9]. The value of our method is further demonstrated when explaining closed-source, proprietary LVLMs on both datasets. In particular, we deploy Claude 3.7[3], Claude 3.5 Sonnet[4] and Claude 3 haiku[5] as classifiers, prompting them to classify given images accordingly. In the case of Claude 3.7, experimentation is conducted with and without thinking. Self-consistency [41] is also employed to guarantee robustness of finally predicted labels, since hallucinations may be present in the LVLM-as-classifiers case. To this end, we repeat the classification process 7 times per image, considering the labels of each run as an indicator of each model's intrinsic classification ambiguity. Finally, we obtain the final label via majority voting.

**Generative Module**   We leverage Stable Diffusion v1.5 Inpainting in the core of the generation process. The proposed edits are first passed through a pipeline of GroundingDINO [28] and SAM (Segment Anything Model) [24] to generate the concept masks. For inpainting, the positive prompt is decided by each edit from $E$, while we also make use of a negative prompt to facilitate image manipulation and enhance realism. Only the masked area and a small expansion around it are affected, so that cohesion of the editing process is encouraged and common generation pitfalls are reduced. This process is repeated until label flip is achieved or the edit set $E$ is exhausted. All experiments are conducted on an L40S GPU (48GB) with an average memory usage of 70% (33.6GB). Technical details are provided in App. **C**

**Evaluation**   Following Jeanneret et al. [23], we evaluate our counterfactual examples (CEs) along three complementary dimensions. Realism is quantified with Fréchet Inception Distance (FID) [16] and the more robust CLIP Maximum Mean Discrepancy (CMMD) [20]. We also evaluate through SimSiam Similarity between the two generated domains [21]. Effectiveness is captured by the Success Rate (the fraction of CEs that flip the classifier) and by the mean number of semantic edits $|E|$ required to reach the target class, a cost-related metric overlooked in earlier CE work [9]. Stability is estimated from the proportion of identical labels obtained across seven independent model runs. Finally, because counterfactuals are meant for human interpretation, we conduct a user study in which volunteers inspect the original image followed by the edited sequence, mark the step where they believe the label should change, and judge whether the final image appears free of noticeable artifacts. More details are provided in App. **D**

---

[2]As we do not consider inter-concept edges, VG-Dense subset proposed in the same paper does not provide any new insights.

[3]us.anthropic.claude-3-7-sonnet-20250219-v1:0

[4]anthropic.claude-3-5-sonnet-20241022-v2:0

[5]anthropic.claude-3-haiku-20240307-v1:0

**Comparisons**  Our primary goal is mostly to explore the explanatory gap between humans and models, harnessing interpretable semantics in a black-box manner rather than proposing a new editor. To verify the framework's potential, we also compare with both SoTA approaches for counterfactual image generation (STEEX [19], DiME [22], ACE [21], TIME [23]), as well as semantic-based counterfactual algorithms that do not generate images [8, 9]. In both cases, we compare with the same datasets utilized by these works, i.e. BDD100K and Visual Genome (Random split) respectively. Of the prior work, STEEX conditions on semantic masks and ACE uses a binary mask of the difference between the explanation and input image for refinement. We do not utilize masks for training, but only for object recognition and editing.

### 4.1  Results on BDD100K

Table 1 summarizes the outcomes of multiple counterfactual generation approaches evaluated on BDD100K. These methods are categorized by the extent of their access to the classifier, whether they require training, and whether they rely on an optimization approach to produce counterfactual images. This distinction is crucial, as white-box methods are trained on the dataset and utilize an optimization strategy which, as evidenced by the results, allows them to achieve nearly flawless performance, producing high-quality counterfactual images of exceptionally high SR. Nevertheless, since they depend heavily on the classifier and dataset, the modifications they make are often subtle and users may struggle to comprehend why the class changes, undercutting the core objective of explaining the classifier. In scenarios where the edits are more overt (e.g., not merely introducing imperceptible noise that shifts the class), these methods still provide no guarantees about the classifier's semantic reasoning, leaving the interpretation of the results highly variable. TIME likewise embeds latent inter-class differences to craft counterfactuals, yet its classifier's vague semantic level still clouds interpretation. V-CECE instead fixes the semantic level and does not require dataset-specific training, letting us ask: *"Does the classifier reason like a human?"* For CNNs (DenseNet, ConvNeXt) the answer is **No**, as SR falls to 84.8–88.9% and the image quality as denoted by FID and CMMD degrades heavily as the number of steps increase. However, this should not be seen as a flaw in V-CECE itself, as its distinct behavior when using LVLMs as classifiers achieves near-perfect SR and even outperforms TIME—despite the latter being trained on the dataset—in generating higher-quality images. Consequently, any attempt to explain CNN classifiers at a human-like semantic level may result in poor image quality at best, and potentially misleading outputs at worst.

To validate the above hypothesis, we compare the CNN-based DenseNet with the LVLMs as classifiers not only in terms of visual metrics, but also by juxtaposing their average number of semantic edits $|E|$ needed for label flip (Avg$|E|$ column). We observe a significant disparity in both the avg $|E|$ and the visual metrics between DenseNet and LVLM classifiers. Without relying on the previous analysis and metrics, and by considering only the number of edits, this discrepancy could arise from either differences in the level of explanation required by DenseNet or the need for more semantic edits to alter its classification. Among the CNNs, EfficientNet flips labels fastest, requiring markedly fewer optimization steps. We attribute this to its leaner architecture: a smaller computational footprint and compound scaling constrain activations to semantically pertinent regions, reducing drift into irrelevant areas during training [4]. This suggests that architecture can play a role in bridging the semantic gap without sacrificing fidelity, but neural classifiers still lag behind LVLMs.

A finer-grained comparison of Claude 3.7 Sonnet *with* versus *without* the thinking module underscores the role of prompting. Activating the module demands more edits to change the label and yields lower FID scores, even though the underlying weights are identical. This aligns with reports that Chain-of-Thought can hamper visual recognition tasks, likely because verbal reasoning struggles to capture visual cues [27].

**Human evaluation**  The human survey sheds some light to the underlying cause of the human-model explanation level discrepancy. The related findings are detailed in Table 2 (more results in App. **E**), where the average number of edits by both the respective models and humans on the same annotated BDD subset is presented. These results suggest that, from a human perspective, the label change in DenseNet should ideally occur **three edits earlier** on average[6] than currently necessary. This points to a potential misalignment between human judgments and classifier perspectives on this task. Although this discrepancy does not necessarily indicate that DenseNet operates at a different

---

[6]Averaging numbers for all ordering methods per classifier.

Table 1: Comparison of CE image generation methods across metrics and model design choices. (-) denotes that related results are not reported by authors. **Bold** indicates best black-box results.

| Method | FID ($\downarrow$) | CMMD ($\downarrow$) | S3 ($\uparrow$) | SR ($\uparrow$) | Avg.$|E|$ ($\downarrow$) | Access | Training | Optimiz. |
|---|---|---|---|---|---|---|---|---|
| STEEX | 58.8 | – | – | 99.5 | – | white-box | days | ✓ |
| DiME | 7.94 | – | 0.9463 | 90.5 | – | white-box | days | ✓ |
| ACE $\ell_1$ | 1.02 | – | 0.9970 | 99.9 | – | white-box | days | ✓ |
| ACE $\ell_2$ | 1.56 | – | 0.9946 | 99.9 | – | white-box | days | ✓ |
| TIME | 51.5 | – | 0.7651 | 81.8 | – | **black-box** | hours | ✗ |
| *V-CECE – DenseNet classifier* | | | | | | | | |
| V-CECE$_{\text{Local}}$ | 90.42 | 1.101 | 0.6254 | 88.9 | 4.77 | | | |
| V-CECE$_{\text{Global}}$ | 99.37 | 1.232 | 0.5489 | 85.8 | 5.37 | **black-box** | N/A | ✗ |
| V-CECE$_{\text{Local-Global}}$ | 81.90 | 1.092 | 0.6169 | 84.8 | 5.23 | | | |
| *V-CECE – ConvNext classifier* | | | | | | | | |
| V-CECE$_{\text{Local}}$ | 84.08 | 1.111 | 0.6130 | 95.36 | 4.91 | | | |
| V-CECE$_{\text{Global}}$ | 99.21 | 1.243 | 0.5560 | 81.4 | 5.86 | **black-box** | N/A | ✗ |
| V-CECE$_{\text{Local-Global}}$ | 92.94 | 1.312 | 0.5510 | 82.09 | 5.81 | | | |
| *V-CECE – EfficientNet classifier* | | | | | | | | |
| V-CECE$_{\text{Local}}$ | 67.27 | 0.767 | 0.6950 | 98.07 | 3.76 | | | |
| V-CECE$_{\text{Global}}$ | 69.31 | 0.733 | 0.6940 | 87.82 | 4.1 | **black-box** | N/A | ✗ |
| V-CECE$_{\text{Local-Global}}$ | 69.31 | 0.758 | 0.6930 | 93.03 | 4.01 | | | |
| *V-CECE – Swin classifier* | | | | | | | | |
| V-CECE$_{\text{Local}}$ | 82.93 | 1.027 | 0.6160 | 94.06 | 4.71 | | | |
| V-CECE$_{\text{Global}}$ | 92.76 | 1.025 | 0.5860 | 83.76 | 5.3 | **black-box** | N/A | ✗ |
| V-CECE$_{\text{Local-Global}}$ | 87.93 | 1.034 | 0.6040 | 82.58 | 5.35 | | | |
| *V-CECE – Claude 3 Haiku classifier* | | | | | | | | |
| V-CECE$_{\text{Local}}$ | 56.93 | 0.566 | 0.7667 | 95.14 | 3.19 | | | |
| V-CECE$_{\text{Global}}$ | 59.94 | 0.516 | 0.7646 | 95.64 | 3.13 | **black-box** | N/A | ✗ |
| V-CECE$_{\text{Local-Global}}$ | 55.05 | 0.527 | 0.7528 | 95.17 | 3.21 | | | |
| *V-CECE – Claude 3.5 Sonnet classifier* | | | | | | | | |
| V-CECE$_{\text{Local}}$ | 62.64 | 0.524 | 0.7593 | 96.60 | 3.10 | | | |
| V-CECE$_{\text{Global}}$ | 45.22 | 0.427 | 0.7635 | 97.80 | 2.65 | **black-box** | N/A | ✗ |
| V-CECE$_{\text{Local-Global}}$ | **42.76** | **0.364** | **0.7970** | 98.10 | **2.44** | | | |
| *V-CECE – Claude 3.7 Sonnet classifier (No thinking)* | | | | | | | | |
| V-CECE$_{\text{Local}}$ | 67.78 | 0.565 | 0.7679 | 99.5 | 3.03 | | | |
| V-CECE$_{\text{Global}}$ | 70.65 | 0.620 | 0.7394 | 98.51 | 3.47 | **black-box** | N/A | ✗ |
| V-CECE$_{\text{Local-Global}}$ | 68.17 | 0.529 | 0.7591 | **99.76** | 3.45 | | | |
| *V-CECE – Claude 3.7 Sonnet classifier (Thinking)* | | | | | | | | |
| V-CECE$_{\text{Local}}$ | 73.36 | 0.762 | 0.6165 | 98.2 | 3.78 | | | |
| V-CECE$_{\text{Global}}$ | 79.28 | 0.829 | 0.6490 | 96.41 | 4.37 | **black-box** | N/A | ✗ |
| V-CECE$_{\text{Local-Global}}$ | 73.51 | 0.876 | 0.6750 | 97.73 | 4.07 | | | |

semantic level from humans, this hypothesis gains support from the fact that the 59.7% of the counterfactual images, are **visually incorrect** (contain generation artifacts or defy commonsense), as indicated by DenseNet's visually correct images rate on average. As a result, DenseNet does not flip its label concurrently with human judgments whilst the images appear more visually accurate. It

Table 2: Average human-survey results regarding perception of quality.

| | Avg. $|E|$ **Model** ($\downarrow$) | Avg. $|E|$ **Human** ($\downarrow$) | **Visually correct images (%)** |
|---|---|---|---|
| DenseNet | 5.22 | 2.21 | 59.71 |
| ConvNext | 7.35 | 2.27 | 34.24 |
| EfficientNet | 5.96 | 2.66 | 30.17 |
| Swin | 6.31 | 2.25 | 56.66 |
| Claude-3-Haiku | 2.91 | 1.88 | 69.58 |
| Claude-3.5-Sonnet | **2.19** | **1.33** | **81.20** |
| Claude-3.7-Sonnet | 2.50 | 1.37 | 79.98 |
| Claude-3.7-Sonnet Thinking | 4.33 | 2.69 | 70.01 |

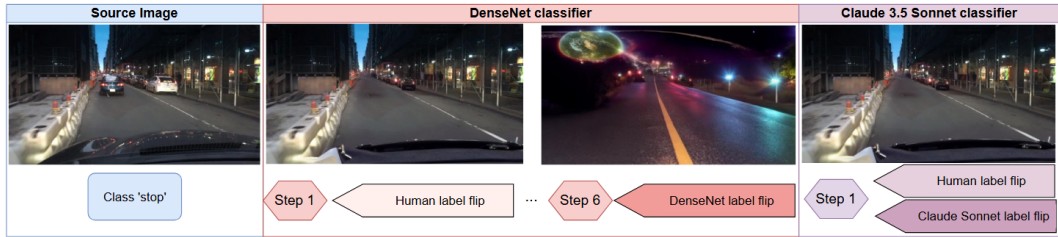

Figure 2: An example of counterfactual generation using the global-local method for two different classifiers on the same input image, including the source image, the intermediate image generated from the edit in step 1 (removal of the car), and the final counterfactual image that prompted a label change. In the case of the LVLM, the counterfactual image is the same as the image from step 1.

does, however, change classes after an average of three additional edits, by which time the 59.7% of generated images start showing artifacts, denoting that their semantic integrity is compromised. Notably, EfficientNet, has the worst performance, indicating that despite the small amount of steps and relatively high fidelity, the images are not interpretable by human standards.

**Qualitative results** Figure 2 presents an example of CE generation using the local-global edit method, displaying the source image from the class "Stop", alongside the outcomes of the steps needed until label flip for DenseNet and Claude 3.5 Sonnet classifiers. DenseNet requires 6 steps of step-by-step generation to alter its label to "Move", leading to a highly edited image: note that the pavement on the left and the building on the right have been unnecessarily removed, while the halo added decreases the correctness of the generation. Inserting such semantically unrelated features raises significant doubts about DenseNet's semantic understanding, as its label flipping relies on such features. The car blocking the way is correctly deleted; however, this edit has been already performed in Step 1, even though DenseNet has not effectively handled this semantic change to alter its label. On the other hand, humans agree that "Stop"→"Move" label flip should happen at Step 1. This perspective correlates with the outcome of Claude 3.5 Sonnet, which requires exactly one step to alter its label. This finding strengthens the assumption that Claude 3.5 Sonnet effectively grasps the semantics of each class, resulting in fewer passes from the generative module.

**Discovering biases** Another interesting observation pertains to the global edits required for a classifier to change its label. By analyzing the edits, we find that for Claude 3.5 Sonnet the most common modifications required to transit from "Stop" to "Move" involve removing the concepts *"car"*, *"pole"*, *"streetlight"*, and *"person"*. Similar are the most common edits for the Claude 3.7 Sonnet with the thinking and not thinking module. For Claude 3 Haiku, the most common concepts are *"car"*, *"vegetation"*, *"pole"*, and *"person"*. In contrast, the edits for the CNN based networks are sporadic, indicating that no consistent steps could reliably alter the class, resorting label-flipping to randomness. To further analyze this finding, we calculate the importance of each concept.

Table 3 presents the importance value of the most prominent concept, along with its standard deviation and the number of concepts with an absolute importance value greater than 0, for each classifier. From this table, it is evident that there is a significant discrepancy between the maximum importance values and the number of important concepts in the CNN-based classifiers compared to the LVLMs. This suggests that CNN-based classifiers tend to modify more objects overall, with less consistency, indicating that there is a randomness in their decision making process.

Table 3: Importance and standard deviation of the most prominent concept for each classifier.

| Classifier | Importance | #(Importance $> 0$) |
|---|---|---|
| DenseNet | $0.23 \pm 0.04$ | 35 |
| Swin | $0.17 \pm 0.03$ | 55 |
| ConvNeXt | $0.16 \pm 0.03$ | 40 |
| EfficientNet | $0.20 \pm 0.03$ | 49 |
| Claude 3 Haiku | $0.23 \pm 0.04$ | 39 |
| Claude 3.5 Sonnet | $0.37 \pm 0.05$ | 31 |
| Claude 3.7 (no thinking) | $0.40 \pm 0.06$ | 28 |
| Claude 3.7 (thinking) | $0.32 \pm 0.05$ | 27 |

These results reinforce that DenseNet's semantic space differs from V-CECE's. Claude 3 Haiku inaccurately links roadside vegetation to the "Stop" label, requiring about one more semantic edit than

Claude 3.5 Sonnet. For Claude 3.7 Sonnet, the critical edits remain the same with and without the thinking module, but their weights fall sharply when thinking is enabled, confirming that the module adds randomness (see Table 1). This global edits analysis underscores the significance of V-CECE, which due to its model-agnostic design, is able to operate even on proprietary models and provide insightful results regarding classification biases. This becomes prominent in the LVLM-as-classifier case, where the semantic levels between the classifier and human explanations align. Despite its popularity in prior CE work, DenseNet yields explanations that are only partially aligned with our human-level semantic annotations. This suggests that, at least for our dataset and metrics, CNN features driven by statistical dependencies may be difficult to translate into clear, concept-level explanations.

**Ambiguity** in deciding the final label can shed more light regarding the classifiers' behavior. For this reason, we extract the label probabilities of the final layer in CNN and transformer-based classifiers, while for LVMLs we keep the prediction for each of the 7 runs; the classification is performed on the source images, as well as on the generated ones for as many steps as required until label flipping. The label probabilities regarding the final class are illustrated in Figure 3, revealing interesting behaviors of the models under scrutiny. Regarding the non-LVLM classifiers, EfficientNet arises as the most consistent model, with label probabilities lying well above the rest. Overall, a downtrend is observed for most classifiers and edit selection strategies, denoting that the more edits are performed, the less confident the classifier is. This is an expected outcome, since artifacts tend to occur when more edit steps are performed, strengthening the requirement for performing as few edit steps as possible. As for LVLMs, The downtrend is less visible in the LVLMs as classifiers case, showcasing an advanced classification robustness despite artifacts in comparison to non-LVLM classifiers, even though some decrease in classification confidence is unavoidable after numerous edit steps.

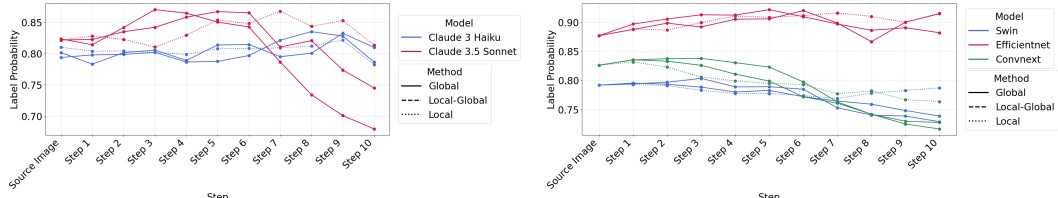

Figure 3: Ambiguity in different LVLMs (left) and CNNs (right) across different stages of the counterfactual generation process.

## 4.2 Results on Visual Genome

Table 4 shows the average number of semantic edits $avg|E|$ required to change class on Visual Genome, compared with previous works on semantic counterfactuals [8, 9]. This comparison reveals that V-CECE not only generates counterfactual images but also produces CEs with significantly fewer semantic edits than the rest. We also report the SR for each classifier employed by V-CECE (SR is not applicable for the non-generative methods we compare with). To evaluate the impact of each edit ordering technique on the generated image quality, we present a related breakdown for all our classifiers in Table 5.

Table 4: Mean count of required edits and SR on VG. **Bold** entries represent the best values.

| Method | ResNet18 | | Claude-3-Haiku | | Claude-3.5-Sonnet | | Generate Image |
|---|---|---|---|---|---|---|---|
| | Avg. $\|E\|$ ($\downarrow$) | SR ($\uparrow$) | Avg. $\|E\|$ ($\downarrow$) | SR ($\uparrow$) | Avg. $\|E\|$ ($\downarrow$) | SR ($\uparrow$) | |
| Dervakos et al. [8] | 12.15 | N/A | 12.83 | N/A | 12.81 | N/A | ✗ |
| Dimitriou et al. [9] | 12.18 | N/A | 12.88 | N/A | 12.84 | N/A | ✗ |
| V-CECELocal | **2.53** | 96.41 | **2.06** | 96.59 | **2.68** | 93.36 | ✓ |
| V-CECEGlobal | 2.54 | 97.77 | 2.74 | 99.49 | **2.68** | 98.13 | ✓ |
| V-CECELocal-global | 2.56 | **98.43** | 2.62 | **99.77** | 2.96 | **98.87** | ✓ |

Table 5: Comparison of methods on **Visual Genome** using **ResNet18** and **Claude** classifiers.

| Method | ResNet18 | | | Claude-3-Haiku | | | Claude-3.5-Sonnet | | |
|---|---|---|---|---|---|---|---|---|---|
| | FID ($\downarrow$) | CMMD ($\downarrow$) | S3 ($\uparrow$) | FID ($\downarrow$) | CMMD ($\downarrow$) | S3 ($\uparrow$) | FID ($\downarrow$) | CMMD ($\downarrow$) | S3 ($\uparrow$) |
| Local | 90.42 | **0.233** | **0.6459** | 78.94 | **0.144** | **0.7154** | 78.94 | **0.212** | **0.6640** |
| Global | 82.05 | 0.295 | 0.6388 | 82.63 | 0.302 | 0.6286 | 90.98 | 0.292 | 0.6195 |
| Local-Global | **81.90** | 0.291 | 0.6169 | **75.66** | 0.302 | 0.6589 | 79.73 | 0.268 | 0.6259 |

Additionally, there is a substantial discrepancy between the global explanations provided by [8, 9], and the ones of V-CECE. Specifically, [8] report a total of 121 edits with non-zero importance for changing the label of images initially classified as "Bedroom" using Claude 3.5 Sonnet. In contrast, V-CECE returns only 12 edits of non-zero importance for the same dataset and classifier, denoting that its global edits proposed after generation are significantly less noisy.

## 5 Limitations and Future Work

We recognize certain limitations within our framework and evaluation. Regarding our human evaluation experiments, the current cohort is modest in size and scope, which limits statistical power, precision, and generalizability; accordingly, results should be regarded as early-insights into a significant problem in AI explainability. To address this, we will broaden the cohort to include participants from varying disciplines and quantify inter-individual variability, for example, how a single sample is interpreted by different human raters, to assess inter-rater reliability and identify sources of disagreement. Apart from enhancing the human survey as a part of future work, we would like to extend the evaluation of the framework across additional disciplines, including the medical domain, to surface challenges and explanation needs in settings with greater field dependence. We also plan to assess robustness to realistic noise in real-world data and measure its impact on both performance and interpretability. In parallel, we will evaluate white-box generative models and examine whether additive bias is detrimental for the editing modules, as well as the effects of masking and segmentation choices. Collectively, these steps are intended to strengthen external validity, reduce uncertainty around effect estimates, and guide design refinements.

## 6 Conclusion

In this work, we present the V-CECE framework which aims to explore the explanatory gap between classifiers and humans driven by semantics. We prove that when employing LVLMs as classifiers we achieve a compatible semantic comprehension with humans, whereas that does not hold for favorable CNN or ViT classifiers utilized in prior literature. Our black-box framework is able to incorporate any classifier for explanation without any training, providing human-level and discrete CEs, the classifier's degree of semantic understanding, and general classification biases. We hope this work proposes a new frontier for explainability analysis, where semantic coherence for artificial intelligence models is at the forefront.

## 7 Acknowledgment

This work was supported by AWS resources, which were provided by the National Infrastructures for Research and Technology GRNET and funded by the EU Recovery and Resiliency Facility.

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

# A LVLMs-as-classifiers prompts

**BDD100K Dataset** The prompt for LVLM-based classification is provided in Table 6, considering 'start' and 'stop' as the {str_categories}. The LVLM is forced to focus on the semantics that define each driving situation, since they are definitive for classification based on concepts.

Table 6: Classification prompt for BDD100K

Classify each image in their appropriate class according to the driving situation they depict. Valid class labels are {str_categories} and only these, depending on whether the car has to move or stop based on its surroundings. You need to classify the images in one of these classes. Pay attention to the semantics that define each class. Return me only the label of the scene depicted and nothing else.

**Visual Genome Dataset** In Table 7 we show the classification prompt used to classify an image from Visual Genome in one of its appropriate categories belonging in the {str_categories} list. The LVLM is forced to focus on the semantics that define each class, since they are definitive for classification based on concepts.

Table 7: Classification prompt for Visual Genome

Classify each image in their appropriate class according to the scene they depict. Valid classes are {str_categories} and only these, so you need to classify the images in one of these classes. Pay attention to the semantics that define each class. Return me only the label of the scene depicted and nothing else.

# B Prompts for performing the edits

As mentioned in the 3.2 section, there are three ways to define which edits are going to be performed and in which order.

In the *local editing* approach, the LVLM serves as the only decision-making module to order the edits produced from the explanation component. In each step, only one edit is selected and passed to the generative component. This assists in performing a small number of steps until label flip, since label flip may occur before the edits proposed in $E$ is exhausted (an assumption that is verified, based on the results of Table 4, in which our generative V-CECE consistently performs fewer edits that its non-generative counterparts). Other than that, performing one step at a time allows for more high-quality generations from the point of the generative component.

The prompt that arranges the local edits at each step is illustrated in Table 8, determining the selection of a $I, D, R$ edit based on its assumed commonsense understanding, which is triggered using a suitable example.

The prompts used by the LVLM to perform the insert and delete edits are provided in Tables 9, 10. This procedure is needed to ensure commonsense of performed edits. At the same time, it assists the mask generator of the generative component to define the object that should appear after deleting another object, effectively handling occlusion, while also masking a suitable area that an existing object spans in case a new object has to be added in relation to it.

# C Generative Component

In our configuration, object detection operates with a confidence threshold of 0.3, guiding the inclusion or exclusion of specific object classes via textual prompts. The bounding boxes around detected objects are expanded by 35 pixels, with a soft boundary applied using a mask blur of 10 pixels. The expansion is required in order for fewer artifacts to emerge from the text prompts, as further contextual information is added and the areas to be modified are restricted.

For inpainting, the process adheres strictly to the provided guidance, with a classifier-free guidance scale of 10, instructing the model to strongly follow the given prompts. A denoising strength of 1

Table 8: Local edits prompt: defined the operations $(I, D, R)$ that are best to be performed in each step, based on the remaining edits and the image.

---

I want to remove some objects and add others. I would like you to find the best possible edit for the image, but I want only a single edit.

You can choose from the following options: - Add an object from the "Add" list. In this case please give the answer in the format: ["add", "added_object", "target where the added object will appear in front of"]. Avoid positional description such as "over", "next to", "above" etc. - Remove an object from the "Remove" list. In this case please give the answer in the format: ["remove", "removed_object", "the object that is behind the object when it is removed e.g. wall, floor, background"].
- Replace an object from the "Remove" list with one from the "Add" list. In this case please give the answer in the format: ["replace", "removed_object", "added_object"].
So, you need to decide whether to add, remove, or replace an object.
For example:
Object list: [couch, lamp, window]
Add list: [bed, curtain, blanket]
Remove list: [lamp, couch]

Step: Replace couch with bed.

Another valid step might be:
Step: ["add", "curtain", "window"].
However, the step ["add", "blanket", "couch"] is not a logical step because the couch is on the remove list. If we put the blanket on the couch, we would still have to remove the couch and thus the blanket as well.
Please respond with only a single step and make the most logical edit you can based on the image I have provided.

Object list: objects
Add list: added_objs
Remove list: removed_objs
Step:

---

Table 9: Prompt defining the addition of objects in the image.

---

I want to add an object in the image. Please specify what is the object that is target where the added object will appear in front of. Avoid positional description such as "over", "next to", "above" etc. Please respond with a single item, without any additional text. I want to parse this answer automatically, so it is crucial to return only a single object without any explanation, or additional text!
For example:
Add: "painting"
Answer: "wall"
Add: "pillow"
Answer: "bed"
Add: obj
Answer:

---

is used, ensuring the inpainted areas undergo full transformation based on the prompt. The Stable Diffusion v1.5 Inpainting model processes the image for 40 steps, using the a DPM++ 2M SDE sampler, with an automatically chosen scheduler. The pipeline uses a default random seed, ensuring reproducibility with the specification of a fixed seed, while no variation seed is applied, preserving consistency in the output. Additionally, a high-resolution fix is enabled, improving the final image quality through a secondary upscaling pass.

Table 10: Prompt defining the deletion of objects in the image.

---

I want to remove an object from the image. Please specify what is the object that is behind the object when it is removed e.g. wall, floor, background. Please respond with a single item, without any additional text. I want to parse this answer automatically, so it is crucial to return only a single object without any explanation, or additional text!
For example:
Remove: "painting"
Answer: "wall"
Remove: "pillow"
Answer: "bed"
Remove: obj
Answer:

---

# D  Qualitative Results

In the following Figures, we present some additional qualitative results as occurring from V-CECE pipeline. Specifically, in Figures 4, 5 we present some successful generations stemming from DenseNet-suggested edits. DenseNet tends to perform more steps on average in comparison to the LVLM classifiers (as analyzed in Table 2), which often leads to misgenerations, as the generative module is unable to handle the complex editing procedure arising as a result of requesting multiple edits in a row. However, in several cases, DenseNet-driven edits lead to successful counterfactual generations, as illustrated below.

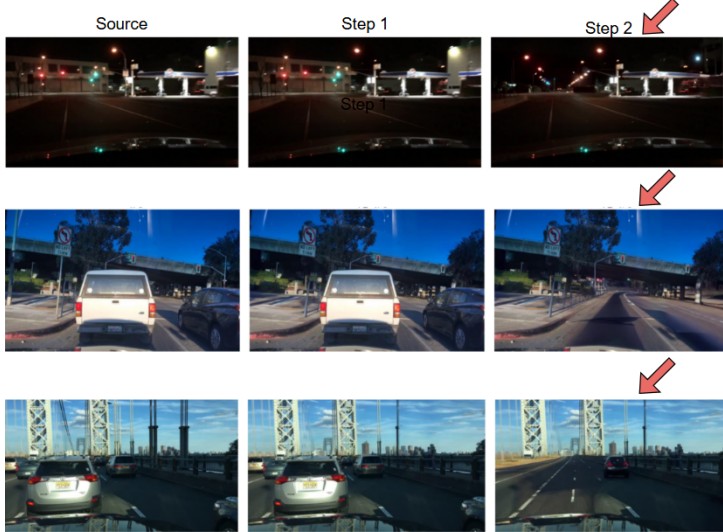

Figure 4: Successful generations after 2 steps of edits for DenseNet classifier. The red arrow denotes the step at which humans perceive label-flipping. In the presented case, DenseNet flips label concurrently with humans and generation terminates.

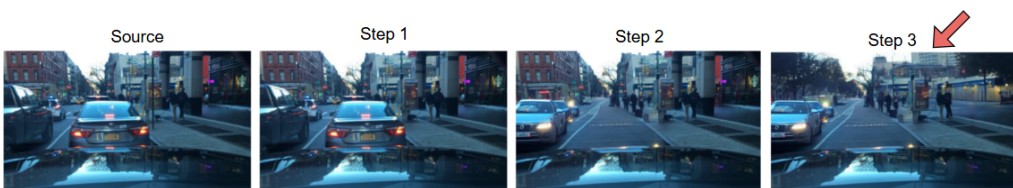

Figure 5: Successful generations after 3 steps of edits for DenseNet classifier. The red arrow denotes the step at which humans perceive label-flipping. In the presented case, DenseNet flips label concurrently with humans and generation terminates.

Interestingly, the success of the performed edit is non-trivial, since removing large objects easily leads to artifacts. Nevertheless, BDD100K images often depict large cars (being close to our point of view from the driver's seat), rendering successful edits challenging. This is a reason why prioritizing the edits in an influential way with respect to the classifier under explanation is crucial.

There are some cases where the classifiers need more steps to identify label-flipping, contrary to humans. Such scenarios are illustrated in Figure 6: classifiers identify label flipping in 3 steps (1st row) and 2 steps (2nd row) instead of one step that a human perceives as necessary. Therefore, the classifier instruct the generation module to proceed, leading to irrelevant edits to the class transition semantics. For example, in the first case, the black car in the same lane as our point of view is removed in Step 1, allowing the transition from "Stop" to "Move" according to humans. The classifier however, cannot perceive this change as influential, concluding to a counterfactual image in which the buildings in the front have been removed, and a black object has been added on the upper right of the frame. However, these changes are totally irrelevant to the queried driving situation. The classifier is probably biased towards certain semantics, or even pixel distributions, therefore being fooled under such transformation, instead of flipping during the removal of the black car in Step 1. In the second case, he white car in the front is removed at Step 1, correctly marked by humans as a "Move" situation. The classifier instructs further generation, resulting in the replacement of the tree with a street light in the front. Nevertheless, this semantic edit is not associated with whether one has to brake or move, deeming this operation as an extraneous edit, wrongly imposed by limited semantic comprehension of the classifier. In the last case, human and classifier perception of label-flipping agree, since the removal of the car in the front suggests transiting to the "Move" class. However, we observe a visual artifact in place of the big car. This example denotes the limitations of the generation module employed in our experimentation, suggesting that even if a single step is performed towards counterfactual generation, it is not guaranteed that the resulting image will be of good quality. Once again, removing large objects is a tough endeavor itself for visual editors, and it is rather unpredictable whether this operation will be performed without any detectable artifact.

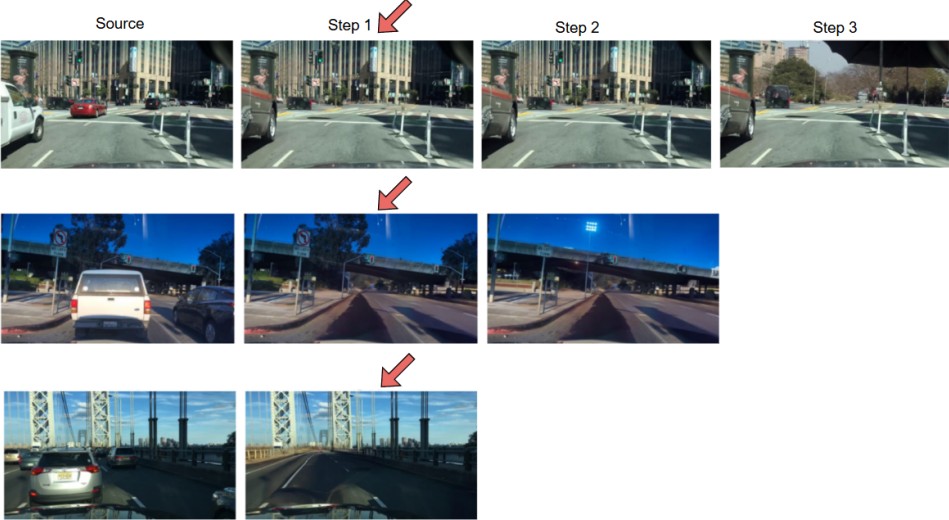

Figure 6: Interesting cases of sub-optimal counterfactual generations. The red arrow denotes the step at which humans perceive label-flipping. In the first two cases, classifiers flip label later than humans; therefore, generation terminates later than necessary. In the last case, humans and classifier perception align, but generation is not devoid of artifacts.

# E   Human survey

Our human survey on BDD100K generated counterfactual images was filled by 31 participants. We gathered no personal information about these evaluators. We used the Label Studio platform for evaluation, allowing us to demonstrate image sequences, along with the required descriptions and questions. Specifically, the participants were provided with a source image and a sequence of numbered generation steps, as occurring from our experiments (we incorporated all classifiers and all ordering techniques). They were then asked to respond to the following:

- The step at which they believe label flip is happening, given that the source class is always "Stop". If label flip did not happen at all in this specific image sequence, they can reply with "None of the above".
- The visual correctness of the image, given the options "Yes" (if the image is visually correct, meaning that it is absent of severe visual artifacts) and "No".

An example of the questionnaire they were asked to fill is presented in Figure 7.

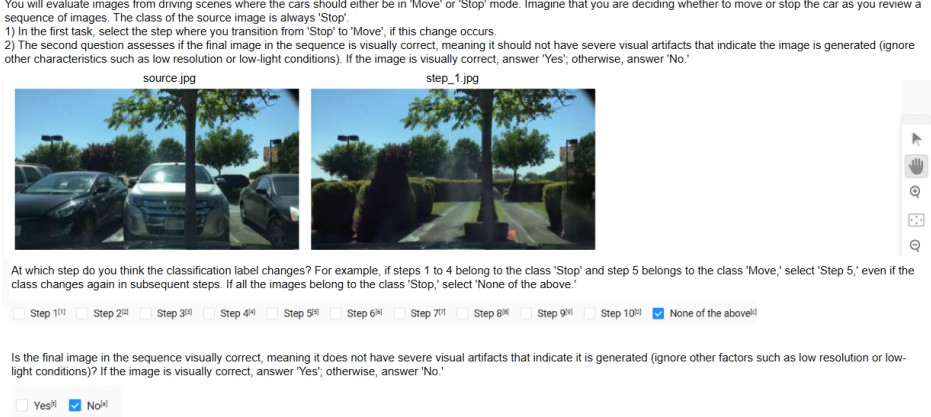

Figure 7: Panel of a human annotation instance in Label Studio.

Consequently, we delve into the human evaluation results, since they are crucial in unveiling the explanatory gap between humans and classifiers via V-CECE explanations. Therefore, we analyze human responses regarding visual correctness (Figures 8, 9, 10) and steps required (Figures 11, 12, 13) until label flip for each ordering method, as well as average values for all methods.

Commencing with DenseNet classifier in Figure 8, its average correctness lies around 60% based on human perception of visual quality. Regarding the ordering techniques for edits, local edits, instructed by Claude 3.5 sonnet on the proposed edit set $|E|$, as occurring from the explanation component, arises as the most successful strategy with 64.58% successful counterfactual generations. The most 'greedy' strategies (with respect to label flipping) that consult global edits score lower, with 57.89% for global and 56.67% for local-global edits.

The local edits are proven as the most successful also in the case of Claude 3 Haiku human results in Figure 9, achieving a 73.47% on visual correctness. On average, Claude 3 Haiku achieves 69.62% correctness indicating a medium agreement with human perception in semantic comprehension for classification.

The patterns changes when Claude 3.5 Sonnet is leveraged as the classifier, where local edits results in only 73.97% correctness, scoring lower than the average of 78.3% on all orderings. Local-global edits lead to 87.88% correctness, the highest percentage overall, suggesting that leveraging model biases in conjunction to LVLM-driven ordering is the best practice for this classifier. Global edits achieve 77.42% correctness, indicating that a 'greedy' edit selection choice is effective, though sub-optimal without proper ordering.

The average number of steps needed for label flip is an informative indicator for the classifier's semantic level as demonstrated on the human survey findings (Table 2). Single-step edits are most

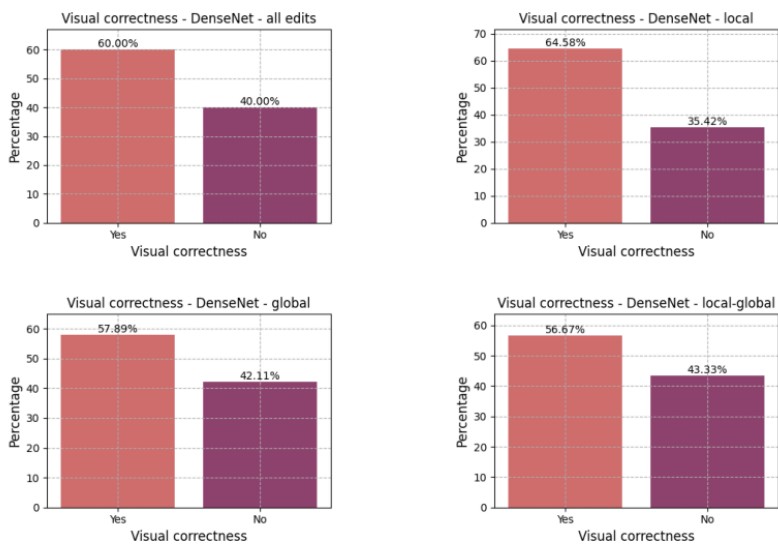

Figure 8: Human evaluation results regarding visual correctness with edits driven from DenseNet classifier.

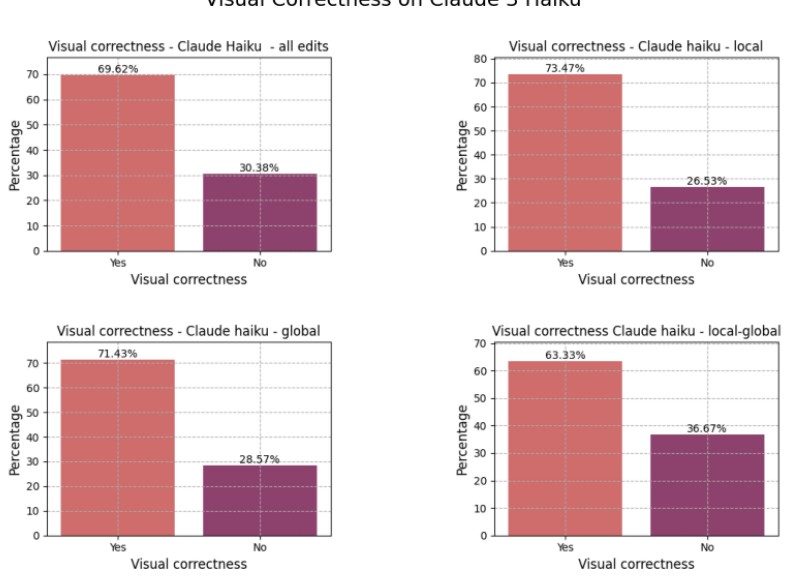

Figure 9: Human evaluation results regarding visual correctness with edits driven from Claude 3 Haiku classifier.

prevalent on average for DenseNet. Interestingly, when local edits are employed, the label-flipping procedure needs two steps as the most frequent step frequency. At the same time, local edits are associated with the best-quality generations for DenseNet, suggesting that despite often needing two steps, the finally generated images are as good as possible, in comparison with other ordering strategies. Furthermore, local and local-global strategies for DenseNet never require more than 5 steps for label flipping, contrary to global edits, which presents few cases of 6 and 7 edits. This finding verifies the effectiveness of Claude 3.5 Sonnet as an edit ordering module, which assists in driving counterfactual generations in fewer steps, thanks to its contextual and spatial understanding.

Regarding Claude 3 Haiku (Figure 12), single-step generations are the most frequent scenario. The behavior of this classifier is more predictable, demonstrating often 2 or 3-step generations, but with a

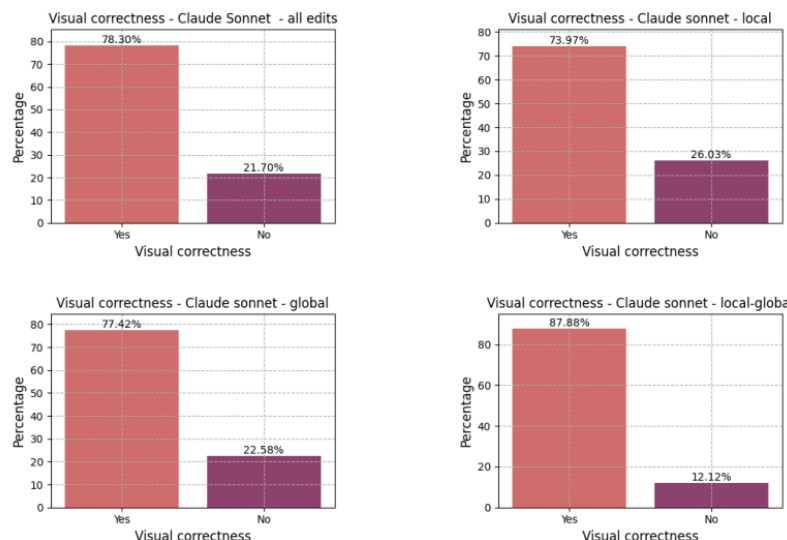

Figure 10: Human evaluation results regarding visual correctness with edits driven from Claude 3.5 Sonnet classifier

striking difference in comparison to the single-step ones. In very few cases, 7 or 8 steps are needed, associated with local and local-global orderings, while for global edits, the steps are at most 5 in few instances. Global edits impose a more aggressive editing strategy towards label flipping, as indicated in Figure 12, but this does not mean these edits are reasonable with respect to the source image semantics, a finding that is cross-verified by the lower image correctness reported previously in Figure 9.

Finally, Claude 3.5 Sonnet presents an outstanding dominance of single-step generations as the most frequent case, as exhibited in Figure 13. Very few cases require more than one step to change classification label and are primarily associated with the local edits strategy (and secondly with the local-global ordering). This verifies that the edits suggested by Claude 3.5 Sonnet in each generation step are suboptimal, agreeing with the visual correctness findings of Figure 10. On the contrary, all generations driven by global edits need only 1 step until label flipping, highlighting this ordering strategy as the most successful one for Claude 3.5 Sonnet classifier, both in terms of editing steps and visual correctness.

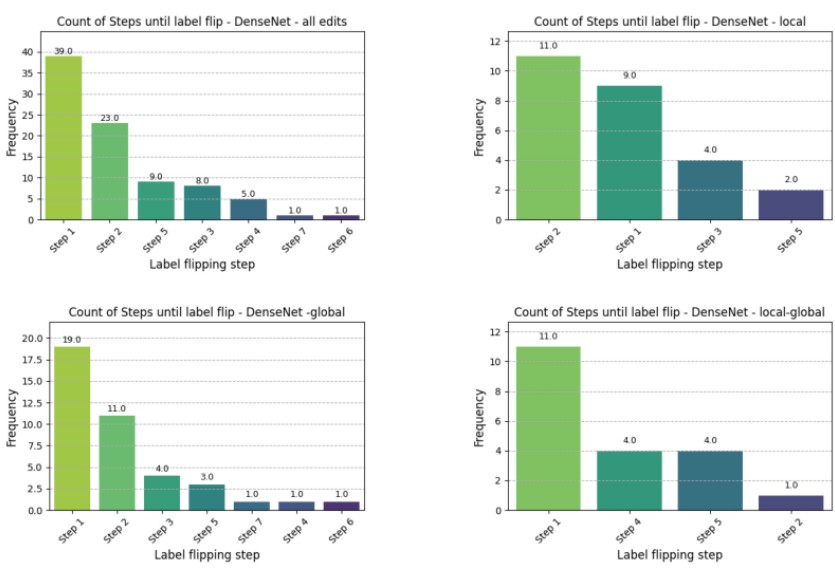

Figure 11: Number of steps until label flip distribution for DenseNet-driven edits.

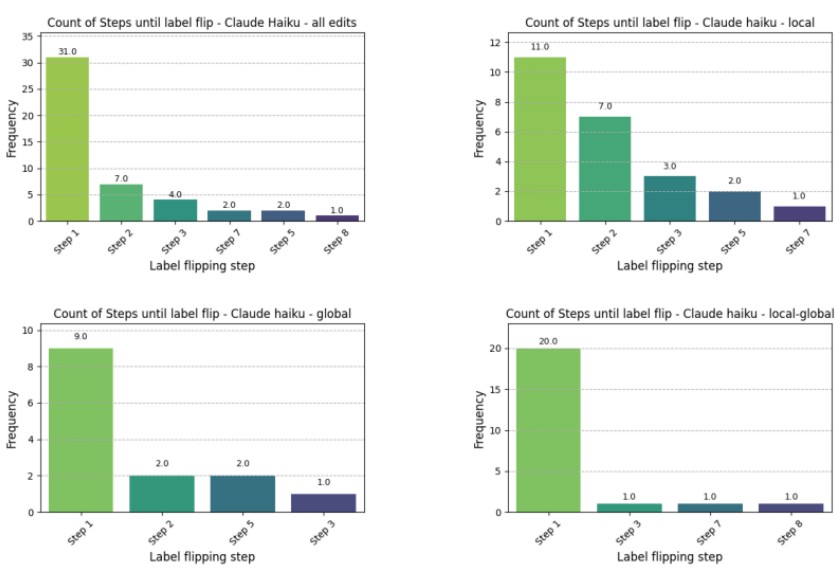

Figure 12: Number of steps until label flip distribution for Claude 3 Haiku-driven edits.

Figure 13: Number of steps until label flip distribution for Claude 3.5 Sonnet-driven edits.

