# OpenReview forum: "V-CECE: Visual Counterfactual Explanations via Conceptual Edits"
_NeurIPS.cc/2025/Conference — NeurIPS 2025 poster_

### Official Review · Reviewer_9F83 · 2025-06-10

**Clarity:** 2
**Significance:** 2
**Originality:** 2
**Rating:** 3
**Confidence:** 5

**Summary:**

The paper introduces V-CECE, a model-agnostic, training-free framework for generating semantically meaningful visual counterfactual explanations by representing image concepts (e.g., objects like “stop sign” or “car”) as nodes in a WordNet-style graph and computing counterfactuals with off-the-shelf diffusion-based models. They then showed that standard vision models were worse compared to LVLMs in terms of visual counterfactual edits that correspond to human-meaningful interpretations of "distance".

**Questions:**

* Can you please describe the user study in detail? So I can assess it properly?
* Why did you only use your method in the user study?
* Can you think of any practical applications for your method? Something with a clear purpose-driven goal?

**Ethical Concerns:**

["NO or VERY MINOR ethics concerns only"]

**Final Justification:**

The paper relies on a user study for the backbone of its evaluation (as it's about human aligned edits), but it was conducted in a somewhat lacking manner in my opinion.

The authors conducted no power analysis, almost no description of the user base, and did not include the full study in their appendix or supplement. Instead results are reported as a basic average, which is (in my opinion) far below the bar of what is scientifically acceptable.

*'Our human survey on BDD100K generated counterfactual images was filled by 31 participants.'* -- This sentence is the sum total of everything we know about the subjects. We don't even know if they were paid, which is a requirement for NeurIPS Checklist item 14.

As such, it is impossible for me to say the study coo-berates their claims and findings, so I can't confidently accept it.

Aside from this I have no issues with the paper.

**Limitations:**

It's ok

**Paper Formatting Concerns:**

It's ok

**Quality:**

2

**Strengths And Weaknesses:**

### Strengths
* I like the model agnostic nature of the framework, although many similar "model agnostic" counterfactual image generation papers exist [1,2], it is a strength that should be acknowledged and I think this approach is quite practical.
* The authors make an attempt to validate this with humans, something often overlooked.
* The focus on human interpretable edits is also quite a nice touch.
* Comparing classic CNNs, ViTs, and LVLMs under the same framework reveals a clear semantic “gap” in how these models represent scenes.

### Weaknesses
* The human study has enormous flaws, it is not described in any detail in the paper, the number of participants, a power analysis, null hypothesis, materials shown, metrics, there is literally almost nothing, not even in the appendix. This is completely unacceptable.
* For me, the clear flaw with this paper is the same as many similar works in the XAI literature, the authors have proposed a new technique, ran it on a few proxy metrics such as num edits required for crossing the decision boundary, and a basic human task of counterfactual simulation, but they have not demonstrated the technique is useful to intended practitioners of the system (whomever that may be).

### Minor points
* The images are all raster, they really should be SVG/PDF
* You cite [5] a lot to justify your arguments, but this work is quite old and never even published (on Google Scholar at least)
* The claim on line 37 that counterfactual images are not reproducible seems bizarre to me, most methods have reproducible code bases.
* You should really start using bookends for your tables to look more professional.

***

[1] Kenny, E.M. and Keane, M.T., 2021, May. On generating plausible counterfactual and semi-factual explanations for deep learning. In Proceedings of the AAAI Conference on Artificial Intelligence (Vol. 35, No. 13, pp. 11575-11585).

[2] Zemni, M., Chen, M., Zablocki, É., Ben-Younes, H., Pérez, P. and Cord, M., 2023. Octet: Object-aware counterfactual explanations. In Proceedings of the IEEE/CVF conference on computer vision and pattern recognition (pp. 15062-15071).

---

> ### Author Rebuttal · Authors · 2025-07-30
>
> Thank you for taking the time to review our submission. Below, we respond to the main concerns raised under the “Weaknesses” and “Questions” sections.
>
> 1. We would like to respectfully disagree with the claim that the human study “has enormous flaws,” or that important information regarding the human survey is missing, even from the appendix (“completely unacceptable”). We believe these characterizations do not accurately reflect the content of the submission. The points raised, including the number of participants, the annotation process, the materials shown, and the methodology used, are in fact clearly presented in the paper in Appendix E. The section provides detailed information about the setup: it involved 31 annotators; the platform used for the survey (we develop a custom platform using Label Studio); it includes full instructions and example screenshots of the annotation interface; and it describes the exact questions asked. The stimuli and evaluation protocol are also explained. In our view, this information sufficiently documents the human study and is in line with the level of detail expected for an evaluation of this kind. Regarding the absence of a null hypothesis or power analysis; our study is exploratory in nature, aiming to assess semantic interpretability and not designed as a confirmatory statistical test. This is consistent with the way similar human studies have been conducted in [1, 2] that we are directly comparing our work with, where such statistical elements, including null hypotheses and power analyses, were likewise not included. The structure, scope, and reporting of our human evaluation are directly aligned with the format and methodology used in [1, 2], which have been peer-reviewed and accepted at top venues (IJCAI, ICML). We did not design a new or ad hoc procedure; instead, we deliberately followed established, community-accepted protocols. Thus, we do not believe the study contains methodological flaws, and we consider the information reported to be appropriate and complete for the goals it serves.
>
> 2. V-CECE introduces a general method to explain any pre-trained classifier in semantic terms, thereby revealing the semantic gap between human and model interpretation; it is not limited to a specific recipient or domain. However, the classifiers we study, CNNs, ViTs, and LVLMs, are among the most widely used architectures in both research and practical applications. In particular, CNNs and ViTs have served as the backbone of image classification for years, making black-box semantic explanation methods for them both timely and important. Similarly, LVLMs are increasingly deployed across domains due to their strong performance and generality, yet they remain opaque by design, warranting targeted explanation endeavors. To this end, V-CECE is designed to be model-agnostic: practitioners can plug in any model to obtain semantic explanations and generate counterfactual images. Beyond previous work, V-CECE provides a novel insight: we uncover and analyze the explanatory gap in generated counterfactuals, highlighting a key mismatch between model behavior and human-understandable semantics. We also include a real-world use case, autonomous driving, where semantic transparency is critical. For example, in a system tasked with deciding whether a vehicle should stop or go, semantic failure could lead to unsafe behavior. Our results show that some classifiers (e.g., DenseNet) may fail in this setting due to limited semantic perception. This finding underscores the importance of deploying semantically-aware models in safety-critical applications and may inform future practices in the design of explainable autonomous systems.
>
> Minor Points
>
> 1. We will change this in the camera ready, should the paper be accepted.
>
> 2. We cite [5] as a theoretical claim of the importance of semantics in explainability. However, this claim has been proven in practice: the works of [1, 2] with which we compare with, are based on semantic counterfactuals, supporting the claim that semantics are crucial for explainability. These works have been published in top-tier conferences (IJCAI, ICML) verifying the validity of this claim. It is also a highly influential paper in the domain of explainability in other works, published in CVPR and ECCV as well [3], [4]. The human study (machine teaching) presented in [2] reveals that semantics are not only important but often adequate to explain differences between classes according to humans. Therefore, there is some extra evidence regarding the importance of semantics in explainability apart from [5], which is once again verified in our work, since we reveal that given a semantic edits algorithm, some classifiers disagree with humans regarding the label-flipping step.
>
> 3. This is not what this sentence attempts to point-out; we do not claim that prior works do not have sufficient code bases. Instead, we mean that they are not reproducible by humans in the sense that a human cannot easily detect what has changed in comparison to the source image when generated edits are dispersed or targeted to non-semantic areas. At the same time, we mention that counterfactual generations are not interpretable; a human cannot accurately reproduce what cannot be interpreted from their point of view.
>
> 4. We will use bookends in the camera-ready.
>
> Questions
>
> 1. All requested details are already included in Appendix E. However, if additional clarification is needed, we are happy to provide it.
>
> 2. Our human study targets to reveal the explanatory gap between classifiers and humans, given different pre-trained classifiers. It does not target to compare generated images between different methods. To the best of our knowledge, no prior work in counterfactual generation makes such a claim. Therefore, there would be no purpose in including generated samples from prior works, since their goal is different. Other than that, this would be unfair for some prior works, since they mostly target label-flipping, but their edits may not be localized, therefore it is harder for humans to assess when label-flipping occurs and why. In this case, the human study would be questionable, as it targets human perception regarding label-flipping; if humans are unable to accurately point the step where the label changes, the results are unreliable, therefore the findings would obscure the semantic level of human understanding. Even if we wanted to directly compare our method to others using a human survey, this would not be possible. The algorithms proposed by [1, 2] do not generate images, but instead provide only theoretical edit suggestions. Similarly, TIME does not support iterative editing; instead, it modifies all relevant areas at once. As a result, it is not possible to isolate and evaluate each individual edit step in a way that would be meaningful for human annotators. Therefore, a direct human-based comparison would be infeasible and potentially misleading.
>
> 3. We presented a practical use-case regarding autonomous driving. Revealing that some popular models, such as DenseNet, diverge from human semantic understanding in this application is crucial and calls for the deployment of semantic-aware models. For example, when engineers would like to deploy a new classifier for a self-driving system, they could check its semantic awareness using VCECE as part of a reliability check pipeline, and compare it to well-known classifiers used in our study (for example, is the new classifier closer to Claude 3.5-the best performer- or closer to CNNs-worst performers- regarding the number of edits?). The reported degree of semantic awareness would suggest improvements in model architecture, if it is found to be non-satisfactory.
>
> We hope these clarifications have addressed the reviewer’s comments and helped resolve any concerns regarding the paper. Thank you once again for your time and effort in reviewing our submission.
>
> [1] Dervakos, Edmund et al. Choose your data wisely: A framework for semantic counterfactuals. In Proceedings of the Thirty-Second International Joint Conference on Artificial Intelligence (IJCAI-23), pages 382–390..
>
> [2] Dimitriou, Angeliki et al. Structure your data: Towards semantic graph counterfactuals. In Proceedings of the 41st International Conference on Machine Learning (ICML 2024), volume 235.
>
> [3] Zemni, Mehdi, et al. "Octet: Object-aware counterfactual explanations." Proceedings of the IEEE/CVF conference on computer vision and pattern recognition. 2023.
>
> [4] Jacob, Paul, et al. "STEEX: steering counterfactual explanations with semantics." European Conference on Computer Vision. Cham: Springer Nature Switzerland, 2022.

---

> > ### Comment · Reviewer_9F83 · 2025-08-07
> >
> > **User Study:** Thanks for the clarifications, in your main paper when you describe the human study (which is really the backbone of the results) you don’t reference the Appendix. As a reader, you can only assume that the authors left out all these details, so this is mostly a misunderstanding, but I am surprised the authors overlooked this. It ties into an overall theme that the paper appears quite rushed, many grammar errors, raster imagery, failing to reference the appendix correctly etc. Admittedly none of this qualifies for rejection in my view, as it's easily fixed for a CRC, but it does signal the submission is perhaps a bit rushed.
> >
> > ***
> >
> > *Beyond previous work, V-CECE provides a novel insight: we uncover and analyze the explanatory gap in generated counterfactuals, highlighting a key mismatch between model behavior and human-understandable semantics. We also include a real-world use case, autonomous driving, where semantic transparency is critical. For example, in a system tasked with deciding whether a vehicle should stop or go, semantic failure could lead to unsafe behavior. Our results show that some classifiers (e.g., DenseNet) may fail in this setting due to limited semantic perception. This finding underscores the importance of deploying semantically-aware models in safety-critical applications and may inform future practices in the design of explainable autonomous systems.*
> >
> > I think the most interesting idea in your paper is about how foundational models seem to exhibit more human-aligned representations of counterfactuals than typical vision specific models, I think that is an idea worth publishing if done correctly. In this case, your human study in theory should be sufficient to show this. However, when I examine the appendix I have more questions than answers. Firstly, 31 participants is definitely on the lower side of what is typically expected in ML, with a number that low, you need to demonstrate the power is sufficient to detect the effect you want (not just show crude averages). Moreover, perhaps more concerning, your appendix only shows a single screenshot of a single counterfactual step on a single question, it is impossible to judge the quality of the study. I tried looking at the Human Survey-submission notebook in the code, but that also lacks these details.
> >
> > ***
> >
> > *We presented a practical use-case regarding autonomous driving. Revealing that some popular models, such as DenseNet, diverge from human semantic understanding in this application is crucial and calls for the deployment of semantic-aware models. For example, when engineers would like to deploy a new classifier for a self-driving system, they could check its semantic awareness using VCECE as part of a reliability check pipeline, and compare it to well-known classifiers used in our study (for example, is the new classifier closer to Claude 3.5-the best performer- or closer to CNNs-worst performers- regarding the number of edits?). The reported degree of semantic awareness would suggest improvements in model architecture, if it is found to be non-satisfactory.*
> >
> > I agree this sounds good in theory, but it is not tested. People often have these theories where their XAI method will be useful, but when actually put to the test, it is more nuanced than that. But I don't think you have to show your study is useful in a real self-driving context like e.g. [1,2], that requires massive recourses beyond the scope of your work,
> >
> > Can you address my points about the user study above? And provide the full study? If that alleviates my concerns, I will raise the score.
> >
> > ***
> >
> > [1] Marcu, A.M., Chen, L., Hünermann, J., Karnsund, A., Hanotte, B., Chidananda, P., Nair, S., Badrinarayanan, V., Kendall, A., Shotton, J. and Arani, E., 2023. LingoQA: Visual Question Answering for Autonomous Driving. arXiv preprint arXiv:2312.14115.
> >
> > [2] Eoin Kenny, Akshay Dharmavaram, Sang Lee et al. Explainable deep learning improves human mental models of self-driving cars, 16 December 2024, PREPRINT (Version 1) available at Research Square [https://doi.org/10.21203/rs.3.rs-5515263/v1]

---

> > > ### Author Response · Authors · 2025-08-07
> > > **Response to Reviewer 9F83**
> > >
> > > We thank the reviewer for their feedback. We respond to the new remarks posed below.
> > >
> > > User Study: We would like to respectfully clarify that the main paper does include a direct reference to the relevant Appendix section in the Ηuman Evaluation paragraph (Line 250). Moreover, we respectfully disagree with the suggestion that the paper appears rushed. Every section of the Appendix is referenced in the main text, and we made a consistent effort to ensure the writing is clear, the structure is coherent, and all necessary details are included. If the reviewer could help clarify certain points of grammatical errors we would appreciate it in order to make the submission more complete for the CR version.
> > >
> > > As for the use of raster images, this is a formatting issue that can easily be resolved during the CRC phase. It was not due to lack of time or oversight but simply a technical choice that will be corrected. We will rectify this, as mentioned in our previous response, for the CR version.
> > >
> > > We also wish to highlight the depth of our contribution. The study includes experiments involving eight different models from three distinct types: CNNs, ViTs, and LVLMs. These models are evaluated on two datasets, with comparisons against seven prior works spanning both the counterfactual vision and explainable AI literature. In addition, we conducted a human study to strengthen and validate our empirical findings. The human survey, thus, provides further insight into the experimental study, showcasing the correlation between our results and human perception.
> > >
> > > Response 1:
> > >
> > > We sincerely thank the reviewer for their recognition that the idea that foundational models exhibit more human-aligned representations of counterfactuals compared to traditional vision-specific models is worth publishing.
> > >
> > > Regarding the human study, we would like to emphasize that “ML applications” is a broad field, and participant counts in human studies can vary depending on the specific task. In our case, a more appropriate comparison is with human evaluations conducted in recent works focused on counterfactual generation, where participant numbers are often in the same range as ours. Specifically, recent papers published at top-tier venues such as ICML,  IJCAI and CVPR report similar participant counts (~30) [1, 2, 3], when such human studies are included at all [4]. Thus, we do not believe that our sample size is on the lower end for this area, but rather in line with current standards in similar works, encouraging fair comparison.
> > >
> > > As for the appendix, the screenshot serves as an example to help readers understand the structure of the survey interface and the type of stimuli shown.  We aimed to strike a balance between providing clarity and keeping the appendix accessible. We would be happy to expand the appendix with additional screenshots from different steps and counterfactuals if the reviewer finds this helpful and more informative. It’s also important to note that our qualitative evaluation is discussed in two dedicated sections, one in the main paper and one in the appendix. In total, we include nine qualitative examples, distributed across both documents, showcasing different models, counterfactual scenarios and step lengths.
> > >
> > > Response 2:
> > >
> > > Testing the application is indeed out of scope for the current study. This practical application was provided as a response to the reviewer’s question. This could be done as future work in a more elaborate setting, precisely in order to review the nuances of implementing a theoretical experiment into an applied one. Our work, at this stage, is an implementation of a highly semantic-aligned task, evaluated in a setting that would be, in the practical stage, transferable. We can add a discussion for this, along with the cited works for clarity, in the camera ready version. We would appreciate it if the reviewer mentioned if this answers their question about applicability.
> > >
> > > We believe that we have addressed the reviewer’s concerns regarding the human study, by clarifying that our participant count is consistent with prior work in counterfactual generation, rather than general ML applications, which vary widely in methodology and scale. We would greatly appreciate further clarification on what is meant by “the full study.” Could the reviewer kindly specify which additional details are expected or which aspects remain unclear, given our two answers during the discussion phase? We would be more than happy to provide any missing information or add further explanation to alleviate any remaining concerns.
> > >
> > > [1] Dervakos, Edmund et al. Choose your data wisely: A framework for semantic counterfactuals (IJCAI 2023)
> > >
> > > [2] Dimitriou, Angeliki et al. Structure your data: Towards semantic graph counterfactuals (ICML 2024)
> > >
> > > [3] Zemni, Mehdi, et al. Octet: Object-aware counterfactual explanations (CVPR 2023)
> > >
> > > [4] Guillaume, Jeanneret et al. Text-to-Image Models for Counterfactual Explanations: A Black-Box Approach (WACV 2024)

---

> ### Comment · Reviewer_9F83 · 2025-08-08
>
> > We would like to respectfully clarify that the main paper does include a direct reference to the relevant Appendix section in the Ηuman Evaluation paragraph (Line 250).
>
> Ah I missed this as appendix is abbreviated
>
>
>
>
> > If the reviewer could help clarify certain points of grammatical errors, we would appreciate it in order to make the submission more complete for the CR version.
>
> Really just minor things, for example the appendix uses incorrect quotations on line 466, alongside my earlier comments. A careful proofread will be enough for this.
>
>
>
>
>
> > As for the use of raster images, this is a formatting issue that can easily be resolved during the CRC phase. It was not due to lack of time or oversight but simply a technical choice that will be corrected. We will rectify this, as mentioned in our previous response, for the CR version.
>
> Thanks
>
>
>
>
>
>
> > Regarding the human study, we would like to emphasize that “ML applications” is a broad field, and participant counts in human studies can vary depending on the specific task. In our case, a more appropriate comparison is with human evaluations conducted in recent works focused on counterfactual generation, where participant numbers are often in the same range as ours. Specifically, recent papers published at top-tier venues such as ICML, IJCAI and CVPR report similar participant counts (~30) [1, 2, 3], when such human studies are included at all [4]. Thus, we do not believe that our sample size is on the lower end for this area, but rather in line with current standards in similar works, encouraging fair comparison.
>
> I think that pointing to other cherry picked studies is a weak argument here, I can also point to other studies which do more rigorous statistical evaluation in the same conferences. My point is not the sample size specifically, but the combination of “low” sample with no power analysis. If you had a sample size of 100+ then effect size will take over (i.e., all p values become significant) and you can just use this as your metric. But for lower samples, you really need to do a proper power analysis and statistical test. Please keep this in mind for future work, I want the field to progress, user testing is important, but we have to do it correctly. Your results seems significant reported as means, but that is not enough for me to say 100%.
>
>
>
> > As for the appendix, the screenshot serves as an example to help readers understand the structure of the survey interface and the type of stimuli shown. We aimed to strike a balance between providing clarity and keeping the appendix accessible. We would be happy to expand the appendix with additional screenshots from different steps and counterfactuals if the reviewer finds this helpful and more informative. It’s also important to note that our qualitative evaluation is discussed in two dedicated sections, one in the main paper and one in the appendix. In total, we include nine qualitative examples, distributed across both documents, showcasing different models, counterfactual scenarios and step lengths.
>
> What I would personally expect to see in a paper like this is the full user study in a PDF I can read and evaluate. I’m sure you understand why, otherwise it is not possible to evaluate it, as I have no idea what the users saw from start to finish. I don’t believe you even took care to ensure a balanced sample between male/female right? I couldn’t find those details. (but perhaps I am wrong)
>
> Unfortunately these are not possible to fix at this point, but I will adjust my score to account for my misunderstandings.

---

> > ### Comment · Reviewer_9F83 · 2025-08-08
> >
> > As a final question however, can I please ask the authors to elaborate on this sentence in the Checklist where you say no IRB approval is needed for human surveys? Did you not get ethics approval?
> >
> > Justification: IRB is not required for human surveys.

---

> > > ### Author Response · Authors · 2025-08-08
> > > **Response to Reviewer 9F83**
> > >
> > > We confirm that no IRB approval was required for the described human survey according to the ethics guidelines of our institution. The survey was fully anonymous and did not involve the collection of any personal or identifiable information from participants (line 350). Specifically, no IP addresses, names, or any other identifiers were recorded, and there was no technical capability to trace which participant provided which specific response. All 31 evaluators took part via the Label Studio platform, where they were only shown generated image sequences and asked to respond to predefined, non-sensitive questions regarding visual correctness and perceived label flips. No demographic, behavioral, medical, or other personal data were collected. Under our institution’s ethics policy, such anonymous, non-invasive, and non-sensitive evaluation tasks do not fall within the scope of studies requiring formal IRB approval. Therefore, the “No IRB approval needed” statement in the checklist accurately reflects the nature of our study and the applicable institutional regulations.

---

> > ### Author Response · Authors · 2025-08-08
> > **Response to Reviewer 9F83**
> >
> > > I think that pointing to other cherry picked studies is a weak argument here, ...
> >
> > We would like to clarify that the studies we referenced are not “cherry-picked,” but rather the works with which we directly compare our approach, following similar survey designs for the same task [1, 2]. One of these is, in fact, the study you yourself suggested for comparison [3]. Our intention in citing these was to provide a relevant baseline from closely related work, rather than referencing general ML applications, which represent a very broad and heterogeneous field. We also included [4] as an example of a method in the same domain that does not conduct any human survey at all, as is the case for many works in this area.
> >
> > We appreciate your advice regarding the inclusion of power analysis and more formal statistical testing for studies with smaller sample sizes. While this was not part of our current work, we will carefully consider this for future studies where it is applicable. Finally, we thank you for acknowledging that our results appear significant, and we are glad that they are convincing for the most part.
> >
> > > What I would personally expect to see in a paper like this is the full user study in a PDF...
> >
> > We understand that providing the full human study could be beneficial for transparency and replicability. While it is not feasible to include the complete study in the appendix-given that it contains approximately 1,000 samples, which would make the appendix impractically long-we will make a PDF with all examples available alongside our code upon publication. This will allow readers to examine the full set of stimuli and responses in detail. The screenshot shown in the appendix is representative of what participants saw: for each sample, a similar layout was presented, but with different images, as shown in the qualitative examples in the main text and the appendix.
> >
> > Regarding participant diversity, we recruited from a varied pool to avoid systematic bias in gender representation and other background characteristics. While no individual demographic information was stored or linked to responses, recruitment procedures ensured that participants shared comparable education levels (all held a higher education degree), domain expertise, and a similar age range, in order to maintain a controlled and homogeneous participant pool.
> >
> > > I will adjust my score to account for my misunderstandings.
> >
> > We thank the reviewer for their willingness to adjust their score and for acknowledging these clarifications.

---

### Official Review · Reviewer_JSSs · 2025-06-24

**Clarity:** 3
**Significance:** 3
**Originality:** 3
**Rating:** 5
**Confidence:** 3

**Summary:**

This paper proposes V-CECE, a black-box, training-free framework for generating visual counterfactual explanations (CEs) through semantically meaningful conceptual edits. The method leverages nearest-neighbor semantic matching and a pre-trained diffusion model to apply optimal edits. It is applied to assess the discrepancy between human reasoning and classifier decisions, including CNNs, ViTs, and LVLMs. The paper emphasizes evaluating models from a human-centered, semantic perspective and offers a novel tool for model auditing.

**Questions:**

**Questions (and Suggestions to Improve Score)**:
1. Why are no modified versions of existing CE generation methods (e.g., TIME, ConceptualCounterfactuals) included as baselines?
Even if exact comparisons are hard, simplified adaptations could offer valuable insights. Including such baselines would significantly strengthen the empirical validation.
2. What prevents integration of components from related works (e.g., Dimitriou et al.) into your pipeline or comparison setup?
Clarifying this would improve the credibility of the claim that no existing methods are comparable.
3. Why is [1] not cited? Their method, like yours, proposes black-box counterfactuals using conceptual edits with minimal changes. How does your framework differ?
4. What exactly is meant by “visually incorrect” images (ln. 256), and how is this judged?
5. Could Figure 1 be simplified or separated into multiple stages for clarity?
6. How does the method handle domains or tasks with weakly defined or abstract concepts (e.g., textures, scenes, medical images)?
7. Why is there no ablation study isolating the effect of the masking step? Since the masking mechanism guides the generative process, it would be helpful to understand its contribution independently.

[1] Conceptual Edits as Counterfactual Explanations (Filandrianos et al., AAAI-MAKE 2022)

**Ethical Concerns:**

["NO or VERY MINOR ethics concerns only"]

**Final Justification:**

I am increasing my score to 5 after reading the rebuttal, as several of my initial concerns were due to misunderstandings that the authors clarified with a detailed and thoughtful response—particularly regarding comparisons to prior work. While an ablation on the masking component and exploration of adapting non-generative baselines for human evaluation would further strengthen the work, these omissions are not critical. The authors' commitment to improving clarity and adding missing details in the camera-ready version further supports the recommendation.

**Limitations:**

Limitations are not sufficiently touched upon.

Additionally:
- Limited to object-level concepts: The reliance on masking and localized edits constrains the method to object-based or spatially separable concepts. More abstract or distributed features (e.g. texture, scene layout) are harder to capture or edit.
- Semantic edit optimality depends on image retrieval: The guarantee of optimal edits depends heavily on the quality and diversity of the dataset used to retrieve the closest image pair. In low-data or biased domains, the optimality guarantee may break down.

**Paper Formatting Concerns:**

- Figure 1 is visually cluttered and lacks an informative caption.
- Typos (e.g., ln. 110: “sj” should be “si”).
- "Visually incorrect" is not well defined (ln. 256).
- Table 3 is not cited in the main text.
- Related work section could better highlight the differences from similar CE generation pipelines.

**Quality:**

3

**Strengths And Weaknesses:**

**Strengths**:
- Tackles an underexplored yet important task: semantic-level evaluation of vision classifiers.
- Training-free and black-box framework with plug-and-play generative capabilities, that supports multiple types of classifiers (CNNs, LVLMs).
- Methodology is explained clearly.
- Incorporates human evaluation to measure classifier-human alignment.

**Weaknesses**:
- No comparisons to existing or adapted CE generation methods like TIME, even in ablation or simplified form. Also, related work that could be used for comparison is omitted [1]
- The setting in prior comparisons (e.g., Dervakos et al., Dimitriou et al.) is underexplained.
- Some parts of the writing and figures lack clarity (e.g., Figure 1, Section 4.1).
- Limited discussion of failure cases and when the method might not work (e.g., ambiguous concept boundaries).

[1] Conceptual Edits as Counterfactual Explanations (Filandrianos et al., AAAI-MAKE 2022)

---

> ### Author Rebuttal · Authors · 2025-07-30
>
> We sincerely thank the reviewer for their thoughtful and constructive feedback. We truly appreciate the time and effort dedicated to deeply understanding the paper and for providing such detailed comments and suggestions.
>
> Weaknesses:
> 1. Comparison with TIME and prior white-box methodologies is provided in Table 1. Regarding the omitted citation [1] (Filandrianos et al., AAAI-MAKE 2022), we would like to clarify that it presents the same core algorithm as in [2], written by the same authors, which we already cite and use for direct comparison. We chose to refer to [2] instead, as it includes more extensive experimentation and a broader evaluation across multiple datasets, making it more appropriate for empirical comparison.
> That said, we appreciate the reviewer’s suggestion and agree that including [1] as a reference could provide helpful additional context. We will add it to the related work section of the camera-ready version and briefly explain its relation to the more comprehensive follow-up work of [2].
>
> 2. We did not elaborate on the evaluation setup in detail, primarily due to space limitations. As noted in lines 208–210, we follow the exact evaluation methodology described in the related works we compare against ([2] and Dimitriou et al. [3]), to ensure a fair and meaningful comparison. A more detailed explanation of the evaluation procedure will be included in the camera-ready version. Both [2] and [3] rely on semantic counterfactuals driven by knowledge graphs. Their methods retrieve counterfactual images and extract semantic edits that indicate what needs to change in order to shift from the source class to the counterfactual class. However, they do not generate the proposed edits, and thus no minimally altered image is returned. For this reason, we compare the methods based on label flip and number of edits performance on the Visual Genome dataset (Tables 4 and 5).
>
> 3. We will revise the parts referenced to improve clarity. If the reviewer could point out specific aspects that were unclear, it would help us focus on our improvements more effectively.
>
> 4. We have placed discussion for limitations in the Appendix Line 504. There are certain cases in which artifacts may exist, despite the mitigation procedure,a known problem with diffusion inpainting and generation in general [4]. There are also certain cases in which the classifier might flip the label sooner than human perception, but this is highly variable. We will update the camera ready with a clear limitations section in the Appendix.
>
> Questions:
> 1. We would like to kindly ask for further clarification regarding what type of adaptation is being suggested. It is not entirely clear to us whether the reviewer is referring to adapting existing methods to operate on new datasets or to modifying their outputs for compatibility with our evaluation pipeline. Below, we provide responses covering both possible interpretations:
> Adaptation of existing methods to other datasets:
> We aim for fair comparisons by evaluating each method in the exact setting proposed by its original authors. We run each baseline using the same dataset, classifier, and data splits as the original work. While adapting each method to a different dataset might seem straightforward in theory, such adaptations often require careful tuning and design choices that may deviate from the original intent. We deliberately avoided introducing such changes to prevent the risk of unfair comparisons.
> Adaptation of methods for human evaluation:
> TIME is included in Table 1 for reference. However, an empirical comparison with human studies is difficult because TIME does not apply semantic edits step-by-step, but instead performs large-scale image manipulations. As a result, it is difficult for human evaluators to identify which change led to the label flip. Additionally, we did not include Conceptual Counterfactuals in the human survey, as it does not generate counterfactual images. Since human evaluation requires visual inspection and judgment, such methods cannot be directly incorporated. If we have misunderstood the reviewer’s concern, we would be grateful for further clarification. We would be happy to provide a more targeted response to ensure that all aspects of the question are fully addressed.
>
> 2. We integrate components from [2] and [3]: we base our edits on semantic distances driven by a knowledge graph. Their main limitation is that they do not generate the counterfactual image that corresponds to the semantic edits. This way, not only do they not provide an actual counterfactual image, but by generation it is possible that we need fewer edits to change the class, indicating that both [2] and [3] propose more expensive edit set. Additionally, both [2] and [3] take it for granted that the classifiers under inspection operate based on semantic concepts, without providing any form of evaluation or verification. In contrast, by generating actual counterfactual images and observing whether fewer semantic edits suffice to flip the prediction, we are able to examine and test this assumption explicitly.
>
> 3. We will cite the work. Similarly to the previous answer, the method does not generate counterfactual images.
>
> 4. A visually incorrect image has generation artifacts or defies commonsense (ie shadows existing without an object to cast it). This characterization is supported from the human survey, in which participants were asked to identify the visually incorrect images during the generation process.
>
> 5. We will simplify the figure.
>
> 6. The framework aims to identify semantic relationships through well-defined object-oriented semantics. For abstract concepts to be supported, the framework would require a different semantic graph that can provide this information. Since V-CECE is agnostic to the specific source of semantic distances, it can work with any suitable knowledge graph, assuming concept coverage is sufficient. We leave this as future work.
>
> 7. In V-CECE, the Editing Module is treated as a black-box within the generative process. While we conduct ablation studies on all other parts of the algorithm, we did not isolate the effect of the masking step specifically. This is because we do not introduce any novelty regarding the parts of the editing module; the novelty lies on the usage of the module to generate counterfactual images, driven by the novel conceptual counterfactual algorithm. Also, this component has already been indirectly evaluated in the original works (GroundingDINO, Segment Anything Model, Stable Diffusion Inpainting). In those studies, the effect of the masking mechanism on the quality and accuracy of the generated outputs has been analyzed as part of the editing system. We rely on these evaluations for the low-level behavior of the generator, focusing instead on the design and evaluation of the full counterfactual explanation framework. If the reviewer considers this ablation important for evaluating V-CECE, we would be happy to include an analysis.
>
> Limitations:
>
> 1. We recognize this limitation; however, V-CECE can function on a robust enough semantic graph that contains objects. As long as that graph exists for abstract concepts, the framework can extract information. At this stage, this is left as future work.
>
> 2. We would like to clarify that the guarantee of optimality in V-CECE is defined relative to the dataset. V-CECE always returns the shortest semantic edit path within that space, regardless of whether the dataset is biased or sparse. In that sense, the guarantee is not invalidated by dataset limitations. In sparse datasets, the absence of intermediate samples can significantly hinder the construction of interpretable counterfactual transitions.  In such cases, concept transitions become coarse, making it harder to capture smooth, interpretable paths. Prior works [2], [3] rely heavily on the dataset structure, and the absence of intermediate examples can lead to unstable or misleading explanations. However, V-CECE directly addresses this limitation by iteratively generating intermediate counterfactuals, one semantic edit at a time, allowing us to densify the semantic space. This gradual construction helps bridge distant concepts through concrete edits, as shown in Table 4, where V-CECE requires fewer steps to reach a decision boundary.  As for bias, we believe this is where V-CECE provides a key advantage. In [2], [3], biased co-occurrences in the dataset are reflected directly in the explanations, regardless of whether they are causal for the model’s predictions. V-CECE breaks this dependency by applying and validating each semantic edit in isolation. If a biased concept, e.g a tree, is introduced and does not change the classifier’s output, V-CECE will not mark it as influential. This ensures that dataset bias is not mistaken for model bias. A similar limitation exists in methods like TIME, which learn edit patterns directly from training data.
>
> Paper Formatting:
>
> 1. We will update the figure and the caption for clarity .
>
> 2. We will correct the spelling.
>
> 3. A visually incorrect image has generation artifacts or defies commonsense (ie shadows existing without an object to cast it). We will update the definition in the main paper.
>
> 4. Table 3 is cited in line 283-284
>
> [1] Filandrianos et al. Conceptual Edits as Counterfactual Explanations, AAAI-MAKE 2022
>
> [2] Edmund Dervakos et al. Choose your data wisely: A framework for semantic counterfactuals, Proceedings of the Thirty-Second International Joint Conference on Artificial Intelligence
>
> [3] Angeliki Dimitriou et al. Structure your data: Towards semantic graph counterfactuals. Proceedings of the 41st International Conference on Machine Learning,
>
> [4] Cao, Bin, et al. "Synartifact: Classifying and alleviating artifacts in synthetic images via vision-language model." arXiv preprint arXiv:2402.18068 (2024).

---

### Official Review · Reviewer_JVys · 2025-07-03

**Clarity:** 3
**Significance:** 3
**Originality:** 3
**Rating:** 5
**Confidence:** 4

**Summary:**

The authors propose a post hoc, model agnostic, counterfactual explainer for computer vision models. They claim that prior counterfactual frameworks do not adequately address the problem of explaining model decision making processes in terms of how humans understand the model's input features. Their V-CECE method uses a large vision language model to identify how to generate a CF and then uses a diffusion model to actually generate the CF. It may be said that the LVLM is used as a proxy for humans. *How* to generate CFs is decided as follows. Every image has a set of visual semantic concept labels. A counterfactual instance $(x*,y*)$ for an input $(x,y)$ can be found by identifying a set of changes to the set of semantic concepts held by $x$ that lead to $x*$ having semantic concepts characteristic of the class $y*$. Each change incurs a cost. Using these costs as weights, the CF design process can be modeled as bipartite matching with weighted edges. An LVLM is used to identify the *set* of changes to $x$ with a *sequence* of changes to $x$. Each change corresponds to a concept edit. Each concept corresponds to a concept mask. Thus the changes to $x$ can be achieved by using a diffusion model to perform inpainting inside of a concept mask.

The evaluation is carried out quantitatively by measuring realism with FID/CMMD, effectiveness with success rate of CF generation, and stability by the proportion of identical labels obtained across independent runs. V-CECE is compared with several baseline methods from prior literature.

**Questions:**

- I think you did not mean to write $s_i$ twice in lines 109-110?
- This is in response to line 154. How can the authors be sure that the diffusion model they use was not actually trained directly on their data?
- I wonder if the claim on lines 305-308 that explaining DenseNet is pointless because CNNs learn statistical dependencies and as a result do not recover human-level semantics is either too strong in need of more justification. After all, researchers in biology have found correlations between CNN visual processing and human visual processing (at lower levels).

**Ethical Concerns:**

["NO or VERY MINOR ethics concerns only"]

**Final Justification:**

The authors gave a satisfactory response to my small critique. After reviewing their exchange with 9F83 about their user study I have nevertheless decided to retain my scores. I feel a change in XAI where XAI papers must include user studies, and must design user studies according to best practices. However, it was not that long ago that XAI papers could be accepted at top CS conferences without user studies and so I am compelled to be charitable during this time, which I identify as phase where the incorporation of user studies is *becoming* the norm and researchers with CS training are adapting to this new landscape.

**Limitations:**

yes

**Quality:**

3

**Strengths And Weaknesses:**

## Strengths

- V-CECE compares favorably to the baseline methods on the chosen metrics
- The **Discovering biases** subsection in section 4 shows evidence of differences in the semantic space of DenseNet and V-CECE. This is a neat result.
- V-CECE requires the execution of fewer edits than competing methods
- The paper has a human evaluation study. It showed evidence that DenseNet is doing too many edits from a human perspective.

## Weaknesses

- I would have liked more detail in section 3.3 about how the diffusion model executes the edits, or at least a sentence saying that more detail can be found in the **Generative Module** subsection in section 4.

---

> ### Author Rebuttal · Authors · 2025-07-30
>
> We sincerely thank the reviewer for their thoughtful and constructive feedback. We truly appreciate the time and effort dedicated to deeply understanding the paper, which is evident not only in the valuable comments provided but also in the remarkably concise and accurate summary that effectively captures its essence and demonstrates a clear grasp of its core ideas and contributions.
>
> Below, we provide brief responses addressing the points raised under the “Weaknesses” section.
>
> 1. We thank the reviewer for this helpful observation. We agree that additional clarification is needed regarding how the diffusion model executes the edits. Due to space constraints, we had originally included the relevant information in Appendix Section B. However, we fully acknowledge its importance for the reader’s understanding. In the camera-ready version, given the extra space provided, we will incorporate a clear and self-contained explanation of the generative process around lines 192–193, previously available only in the appendix. This will ensure that the underlying mechanism is properly conveyed within the main text.
>
> Responses to Questions:
>
> 1. The second s_i should be s_j, we will fix it for the camera-ready version.
>
> 2. The Stable Diffusion checkpoint utilized for the generative task, as per the description of the model card, has been finetuned for inpainting on laion‑aesthetics v2 5+ , as noted in the model card of the checkpoint on HuggingFace, which does not include images from BDD100K or Visual Genome. The original checkpoint also has been trained on the LAION dataset [1]. Both datasets are from the Common Crawl space, which consists of web page documents and images linked with URLs. As such, the dataset does not include curated datasets, which are invisible to the crawl procedure.
>
> 3. The statement in lines 305–308 refers to the DenseNet classifier utilized in prior work and was made with a specific motivation that we now recognize could be explained more clearly. Prior work using the same classifier (DenseNet using BDD100K), such as [2], reports an inability to generate meaningful counterfactual images (Section 4.3, Limitations), attributing this issue to limitations of their algorithm and to the complexity of the dataset. While we acknowledge that these factors may contribute, we argue that the classifier itself also plays a crucial role, which is typically overlooked in previous studies. Through our analysis, we showed that the model’s perception is not aligned with human-level concepts, which significantly limits the potential for meaningful interpretation. In other words, even with a perfect explainability method, the classifier may still fail to produce meaningful outputs, often resulting in outcomes ranging from visually incorrect images to entirely misleading explanations. We will revise the relevant section to articulate this point more clearly and reinforce the reasoning behind our claim.
>
> Once again, we would like to thank the reviewer for their review, and specifically for the time and effort spent to deeply understand our work. Your thoughtful engagement truly means a lot to us and has helped us improve the quality of our work.
>
> [1] Rombach, Robin, et al. "High-resolution image synthesis with latent diffusion models." Proceedings of the IEEE/CVF conference on computer vision and pattern recognition. 2022.
>
> [2]  Guillaume Jeanneret, Loïc Simon, and Frédéric Jurie. Text-to-image models for counterfactual explanations: a black-box approach. In Proceedings of the IEEE/CVF Winter Conference on Applications of Computer Vision (WACV), January 2024.

---

> > ### Comment · Reviewer_JVys · 2025-08-07
> >
> > First, thanks to the authors for responding to my critique. I am satisfied with the authors' responses to each of the points I raised.

---

### Official Review · Reviewer_zsNj · 2025-07-05

**Clarity:** 3
**Significance:** 4
**Originality:** 4
**Rating:** 4
**Confidence:** 3

**Summary:**

the paper introduces a blackbox visual counterfactual explanation generation framework aimed towards generating meaningful and human interpretable edits to the given picture so that its label flips. the key idea is to first identify a set of possible semantic edits to the image, apply them in sequence in terms of their "cost" of edit, and stop when the minimal cost edit which flips the label is encountered. for identify set of edits, the paper utilizes WordNet which uniquely has "part-of" and "is-a" kind of relations required in this setting, and use the shortest path metric as a proxy for semantic cost.

**Questions:**

Besides the list of weaknesses above, I have the following questions:

1.  How sensitive is the method to the quality of the semantic annotations? If the concets are inferred noisily or incompletely, does it still work?

2. can the method handle unseen concepts outside the annotation vocabulary? if so, how? please provide details.

3. it may also be possible that the edits that actually flip the class label are not annotated in the image, for example a general bright background may be able to classify the image as a "daytime picture". what kind of counterfactual editing can this model generate in such cases?

**Ethical Concerns:**

["NO or VERY MINOR ethics concerns only"]

**Final Justification:**

While I was quite positive to begin with, at the end of the discussion phase, it seemed like there some critical parts still missing in the paper. Especially the experiments that I had requested for with different knowledge graphs is critical to justify the generic claims made in the paper. Therefore I would like to reduce my rating slightly.

**Limitations:**

Authors have given discussion of limitations and potential negative societal impact.

**Paper Formatting Concerns:**

None.

**Quality:**

3

**Strengths And Weaknesses:**

__Strengths:__

1. Tackles an important and critical question in explainable AI research

2. Proposed technique is fully blackbox, generalizable (within the WordNet concept space), and clearly interpretable by humans

3. Evaluations are over different datasets and human evaluations clearly demonstrate the utility of the method.

__Weaknesses:__

1. Authors argue that edits produced by diffusion based counterfactual models are "merely responding to changes in pixel distribution". While it may be true, the proposed method is not entirely safe either. V-CECE proposed edits still rely on annotations and concepts inferred from pixel distributions (e.g., by object identifiers). Thus, despite the valid goal of aligning edits with human semantics, it can not fully eliminate the influence of low-level pixel patterns on classifier decisions. Please discuss this aspect more explicitly.

2. The uniqueness of optimal edit set is not guaranteed. Although it is not a major limitation of the work, it may result in more edits than actually required, thus leading to a potentially weak explanation. This issue could be acknowledged and its impact can be discussed further.

3. Experiments with semantically diverse set of datasets is critical for strengthening the claim of generalizability of the proposed model.

4. No details of human participants involved is provided.

5. There is a strong dependence on a specific kind of knowledge graph -- it should have (only?) visually groundable relationships organized in a hierarchical manner (e.g., "part-of" or "is-a") for the proposed model to make sense. Instead of just saying "knowledge graph", potentially leading to ambiguities (since Yago or WikiData are also knowledge graphs but quite useless in this task), authors should explicitly list the desiderata from the conceptual graph model that could be used.

6. the proposed model may not be able to handle cases where classifier actually relies on unannotated, global or non-object features (such as brightness, texture, or some other scene-level cue).

---

> ### Author Rebuttal · Authors · 2025-07-30
>
> We sincerely thank the reviewer for their thoughtful and constructive feedback. We truly appreciate the time and effort dedicated to deeply understanding the paper and for providing such detailed comments and suggestions. We are also especially grateful for the kind recognition of the strengths of our work, as well as for highlighting its originality and significance as excellent.
>
> Below, we provide brief responses addressing the points raised under the “Weaknesses” section.
>
> 1. Influence of pixel-level distributions in the proposed method is a very interesting point. In our work, we have used the annotations provided by humans using datasets that include this type of information. However, the same method can also operate using an automated system for extracting annotations, such as an object detector. In this case, indeed, there is an influence from pixel distributions. However, this influence is not directly related to the model under inspection, but rather to the auxiliary model that extracts the objects. For the inspected model, only human-level semantics are used, and the biases or pixel-level dependencies are not explicitly passed on. That said, it is correct that the influence cannot be entirely eliminated unless object detectors improve significantly. We will include a relevant discussion in the final version of the paper. Thank you once again for this interesting observation.
>
> 2. Uniqueness of the optimal edit sets is another compelling point. Even though uniqueness is not guaranteed (e.g., multiple edits of equal cost may exist), this does not result in more edits, as that would imply a higher overall cost. The determinism of the underlying counterfactual algorithm ensures that ambiguous or redundant edits are avoided. In cases where alternative edit sets exist, the pipeline proceeds as usual, and we have no theoretical or empirical indication that this affects the quality of the explanation. Having two different optimal paths does not lead to degradation, as both adhere to the minimal cost criterion.
>
> 3. Diversity of datasets: VG is a highly diverse dataset, covering a wide range of  topics and semantics. Other datasets such as Flickr30k and COCO often depict very similar scenes, without object annotations, however, making additional evaluation on them redundant. BDD, on the other hand, covers a distinct real-world scenario not directly represented in VG, which adds valuable generalization insight to our study. Both datasets also give us the ability to compare with prior work.
>
> 4. Knowledge graph requirements: We use WordNet (line 109), which indeed follows the required hierarchical structure with “part-of” and “is-a” relations. We have not experimented with other graphs, as WordNet already provides satisfactory semantic distances. Nevertheless, V-CECE is not tied to a specific knowledge graph, any conceptual graph that provides semantic distances could be used. We can make this point more explicit in the final version.
>
> 5. Global/non-object features: We acknowledge that the current framework does not capture cases where classifiers rely on unannotated, global, or non-object features such as brightness or texture. However, these are inherently difficult to define in a structured semantic form. For example, brightness is subjective and hard to quantify consistently; even if changed, it does not usually alter the core semantics of the image (e.g., a classroom remains a classroom regardless of lighting conditions). The goal of this work is to explore whether a classifier relies on clearly defined, human-level concepts. Global features fall outside the current scope but present an interesting direction for future work.
>
> Responses to Questions
>
> 1. Sensitivity to annotation quality: The proposed V-CECE pipeline does not explicitly rely on annotation quality but, like most ML systems, benefits from high-quality data. Incomplete annotations may lead to seemingly lower-cost edits if essential concepts are missing. Noisy annotations, on the other hand, could result in more costly or misleading counterfactuals. While the pipeline can operate in such scenarios, data quality remains a critical pre-processing factor.
>
> 2. Out-of-vocabulary concepts: The datasets used in our study include well-known, everyday concepts already represented in WordNet. If V-CECE is applied to a different domain (e.g., medical imaging), the knowledge graph should be adapted accordingly to ensure appropriate coverage. Since V-CECE is agnostic to the specific source of semantic distances, it can work with any suitable knowledge graph, assuming concept coverage is sufficient and existent.
>
> 3. Handling cases like “daytime picture”: Our focus is on well-specified, semantically grounded scenes such as playgrounds, classrooms, and theaters-settings that are easily identifiable by both humans and models. Generic features like “daytime” or “brightness” are not within the current scope. If a classifier is influenced primarily by lighting, that is precisely the type of superficial dependency V-CECE aims to reveal. Style-transfer-like transformations that alter superficial cues without changing semantics could be a future extension to further probe classifier robustness.
>
> Once again, we sincerely thank the reviewer for their time, effort, and thoughtful engagement with our work. Your feedback has been invaluable in helping us improve the quality and clarity of our submission.

---

> > ### Comment · Reviewer_zsNj · 2025-08-04
> >
> > I thank the authors for their responses. While most points are cleared up, a couple of doubts still remain.
> >
> > 1. Would it be too much to request the authors to conduct a small experiment where they introduce noisy annotations and quantify the robustness (or the lack thereof)? (Question 1)
> >
> > 2. it would be important to confirm if the statement "Nevertheless, V-CECE is not tied to a specific knowledge graph, any conceptual graph that provides semantic distances could be used." can be justified with the use of a knowledge graph that is truly a "graph" not a "hierarchy" ? Otherwise, it may not be fair to say "any knowledge graph" when it can not be anything but WordNet.

---

> ### Author Response · Authors · 2025-08-06
> **Response to Reviewer zsNj**
>
> We thank the reviewer for engaging in this conversation and for their thoughtful comments.
>
> Answer to Q1:
>
> Conducting an experiment by introducing noisy labels (such as randomly adding or removing nodes according to a particular distribution) is indeed an insightful idea to extend VCECE findings. We have already started setting up the additional experiments; we deem, however, that for the analysis to be relevant, sufficient experimentation would be required. Therefore, we have concluded that the following experiments are necessary:
>
> Consider different noise types: first, randomly deleting bounding boxes. Secondly, substituting bounding box labels with other labels. In the second case, we should consider substituting with semantically similar objects, semantically distant objects or random objects. Therefore, noise type comprises 4 options.
>
> Experiment with the degree of noise: this corresponds to the amount of bounding boxes to be perturbed within the dataset. We can initially experiment with deleting/substituting 5%, 10%, 15% of bounding boxes to evaluate if there are significant changes in VCECE outcomes. Therefore, noise degree adds 3 options.
>
> Validate the statistical significance of these experiments: in each experiment, the choice of which bounding box is going to be deleted/substituted is random. To eliminate any bias or randomness induced by particularly picking certain labels, we run each experiment 3 times.
>
> Overall, to obtain some meaningful results for one model and dataset, we need 4x3x3=36 runs. At the same time, our analysis involves 8 models per dataset.
>
> In our current experimentation, an end-to-end inference run on VG requires ~72h. Therefore, to run 36 experiments of 72 hours each on average on 4 GPUs in parallel needs ~648 hours for one model and dataset, not to mention an evaluation overhead per model. Unfortunately, even for a single model this experimentation surpasses the given deadline for author-reviewer discussion. Therefore, we will be happy to include this analysis in the camera ready.
>
> In any case, we need to mention that annotation quality is not a direct concern of the proposed VCECE pipeline. On the contrary, as in any machine learning application, the user is responsible for the quality of the data given to any module, and the expected results are analogous to this quality.
>
> Answer to Q2:
> Thank you for raising this important point. Indeed, V-CECE is not restricted to a particular knowledge graph, and we are more than happy to provide additional clarification.
> The knowledge graph is utilized specifically for determining semantic distances between concepts. For example, given two objects, such as a cat and a dog, the semantic distance can be calculated by finding the minimum distance within the graph. If we consider WordNet as our knowledge graph and assign a cost of "1" to each "is-a” relation, the distance between "cat" and "dog" is computed as "4" due to the four intermediate "is-a” edges.
>
> However, the same computation can be conducted using alternative graphs, such as ConceptNet, provided we appropriately define costs for diverse types of relations, such as "part-of," "HasProperty," and others. By assigning explicit costs to these relations, Dijkstra's algorithm can again be applied to compute semantic distances between concepts regardless of the type of graph. Thus, our method is indeed flexible and not exclusively dependent on WordNet or limited to graphs consisting only of "is-a" type relations. We will clarify this explicitly in the camera-ready version of our manuscript.
>
> Thank you once again for your valuable feedback and suggestions, which greatly assist us in enhancing the quality of our work.

---

### Comment · Area_Chair_Gbi8 · 2025-08-06

Thanks reviewer zsNj for engaging in the discussions!

Dear reviewer 9F83, JSSs, and JVys,

Can you look at the author rebuttal and other reviews and engage in the mandatory discussion of this paper please?

AC

---

### Note · Authors · 2025-08-12

Dear Area Chair and Reviewers,

We sincerely thank the reviewers for their engagement during the review process. During this period, we were pleased to gain constructive feeback and we appreciate the effort that went into understanding the submission, evident from the reviews themselves. We are pleased to note that all reviewers have recognized the novelty and significant contribution of our work for the field of explainability and artificial intelligence. During the discussion phase, we have adequately addressed all parts of the submission requested by the reviewers. Namely, these include remarks regarding the effects of the quality of annotations and out-of-vocabulary concepts (Reviewer zsNj), clarifications for the generation process (Reviewer JVys), clarifications on prior work and novelty and the effects of masking on the overall framework (Reviewer JSSs) and clarifications on the human survey (Reviewer 9F83). For the camera-ready version, we will incorporate an analysis on the robustness of the framework on its graph structure as requested, as well as a discussion on the masking effects compared to prior works. We sincerely thank the reviewers for recognizing our contribution and look forward to the final decision.

Sincerely,
Authors

---

### Decision · Program_Chairs · 2025-09-17

**Decision:**

Accept (poster)

**Comment:**

This paper proposes a blackbox visual counterfactual explanation generation framework aimed towards generating meaningful and human interpretable edits to the given picture so that its label flips. the key idea is to first identify a set of possible semantic edits to the image, apply them in sequence in terms of their "cost" of edit, and stop when the minimal cost edit which flips the label is encountered. for identify set of edits, the paper utilizes WordNet which uniquely has "part-of" and "is-a" kind of relations required in this setting, and use the shortest path metric as a proxy for semantic cost.

4 reviewers reviewed this paper. After rebuttal, there were 2 accepts, 1 borderline accept and 1 borderline reject ratings. Reviewers agreed that this paper presents interesting results and addresses an important task. The main negative concerns are around the evaluation, where one reviewer believed that the power of the user study is not enough to support the conclusion this paper makes, and another reviewer concurs with it but suggest leniency because of the general status of user studies in machine learning work. Another reviewer was concerned about whether pixel-level influences would be removed.

Despite the positive scores, AC thinks this is borderline. First of all, this paper did not propose any novel method, what they proposed is a novel approach that utilizes entirely existing modules to perform the task of evaluating the explanation gap between humans and deep networks. This in itself is not a problem, but AC does believe that it puts this paper in the category of "evaluation papers" where the bar on whether the evaluation is conducted fairly and comprehensively would be a bit higher. In this particular case, the user study is conducted on a very biased task of predicting stop/go on BDD100K, which do not cover a broad set of object categories. Besides, as correctly pointed out by the reviewers, a study size 30 is quite small. The authors cited some references, but it should be pointed out that novel algorithms were proposed in all the references the authors cited, whereas this paper only used frozen diffusion models and prompts to generate counterfactual examples. Hence, its main contribution to the field would be the conclusion of running such experiment (e.g. a main claim is that they showed that LVLMs match human preferences better than other deep networks). For that kind of conclusion, we would expect the experiment to be a bit more comprehensive, than only covering 2 classes with limited amount of participants. In fact, Table 5 already shows that the conclusion might be quite different if different categories were covered.

After deliberation, AC decides to accept this paper, for the purpose that this can be considered analogous to a phase-1 clinical trial. but ask the authors to clearly list the limitations of the user study in the final version.